# Fecal transplant from myostatin deletion pigs positively impacts the gut-muscle axis

Zhao-Bo Luo[1,2†], Shengzhong Han[2†], Xi-Jun Yin[2,3†], Hongye Liu[2], Junxia Wang[2], Meifu Xuan[2], Chunyun Hao[4], Danqi Wang[4], Yize Liu[1], Shuangyan Chang[2], Dongxu Li[4], Kai Gao[2], Huiling Li[3], Biaohu Quan[2,3], Lin-Hu Quan[1*], Jin-Dan Kang[2,3*]

[1]Key Laboratory of Natural Medicines of the Changbai Mountain, Ministry of Education, College of Pharmacy, Yanbian University, Yanji, China; [2]Department of Animal Science, College of Agricultural, Yanbian University, Yanji, China; [3]Jilin Provincial Key Laboratory of Transgenic Animal and Embryo Engineering, Yanbian University, Yanji, China; [4]College of Integration Science, Yanbian University, Yanji, China

**\*For correspondence:**
lhquan@ybu.edu.cn (L-HuQ);
jdkang@ybu.edu.cn (J-DanK)

[†]These authors contributed equally to this work

**Competing interest:** The authors declare that no competing interests exist.

**Abstract** The host genome may influence the composition of the intestinal microbiota, and the intestinal microbiota has a significant effect on muscle growth and development. In this study, we found that the deletion of the myostatin (*MSTN*) gene positively regulates the expression of the intestinal tight junction-related genes *TJP1* and *OCLN* through the myosin light-chain kinase/myosin light chain pathway. The intestinal structure of $MSTN^{-/-}$ pigs differed from wild-type, including by the presence of a thicker muscularis and longer plicae. Together, these changes affect the structure of intestinal microbiota. Mice transplanted with the intestinal microbiota of $MSTN^{-/-}$ pigs had myofibers with larger cross-sectional areas and higher fast-twitch glycolytic muscle mass. Microbes responsible for the production of short-chain fatty acids (SCFAs) were enriched in both the $MSTN^{-/-}$ pigs and recipient mice, and SCFAs levels were elevated in the colon contents. We also demonstrated that valeric acid stimulates type IIb myofiber growth by activating the Akt/mTOR pathway via G protein-coupled receptor 43 and ameliorates dexamethasone-induced muscle atrophy. This is the first study to identify the *MSTN* gene-gut microbiota-SCFA axis and its regulatory role in fast-twitch glycolytic muscle growth.

## Editor's evaluation

This study highlights how the deletion of the MSTN gene in pigs affects the gut microbiota and leads to changes in skeletal muscle growth and function. By transplanting the remodeled gut microbiota from MSTN-deleted pigs to mice, the authors demonstrate the selective hypertrophy of fast-twitch glycolytic muscles. Additionally, valeric acid, a microbial metabolite produced in the gut, promotes skeletal muscle growth by activating the Akt/mTOR pathway via the SCFA receptor GPR43 and has potential therapeutic implications for muscle diseases such as muscular dystrophy and sarcopenia.

## Introduction

The decline in muscle mass is a considerable health problem that reduces quality of life and increases the risks of morbidity and mortality (*Newman et al., 2006*; *Srikanthan and Karlamangla, 2014*). It contributes to the onset of various diseases, such as sarcopenia, obesity, diabetes, and cancer

(*Chen et al., 2021a*; *Silveira et al., 2021*). Myostatin (MSTN), a member of the transforming growth factor β family, is a major regulator of skeletal muscle growth and development (*Chen et al., 2021b*), substantial muscle hypertrophy characterizes animals and humans with *MSTN* mutations (*McPherron et al., 1997*; *Ceccobelli et al., 2022*; *McPherron and Lee, 1997*; *Kambadur et al., 1997*; *Mosher et al., 2007*; *Kang et al., 2017*; *Schuelke et al., 2004*). Recently, various MSTN inhibitors, including monoclonal antibodies, have been evaluated for their potential to treat muscle disorders, such as sarcopenia and cancer-associated cachexia, in clinical trials (*Kim et al., 2021*; *Cho et al., 2022*). Notably, *MSTN* is not only expressed in skeletal muscle, but also in smooth muscle, including in the intestine, and participates in various metabolic processes (*Sundaresan et al., 2008*; *Verzola et al., 2017*; *Esposito et al., 2020*; *Kovanecz et al., 2017*). Previous studies have shown that the mutation of *MSTN* alters the composition of the intestinal microbiota in pigs (*Pei et al., 2021*). However, how *MSTN* deletion affects the gut microbiota remains unclear.

Genetic variations have been shown to affect the composition of the gut microbiota. A mutation in human *SLC30A2* causes a reduction in intestinal zinc transport and greater abundance of *Clostridiales* and *Bacteroidales*, resulting in mucosal inflammation and intestinal dysfunction (*Kelleher et al., 2022*). Moreover, gut *SLC2A1* gene deletion alters the abundances of *Barnesiella intestinis* and *Faecalibaculum rodentium*, promotes fat accumulation, and impairs sugar tolerance (*He et al., 2022*). These results suggest that host genes can influence the gut microbiota, and thereby regulate physiologic processes. Furthermore, surgery-induced changes in the intestinal structure, such as intestinal length, epithelial thickness, and surface area, can affect intestinal function and microbial composition (*Seganfredo et al., 2017*; *Nicoletti et al., 2017*; *Agus et al., 2018*). Barrier defects are accompanied by major changes in the fecal microbiota and a significant reduction in the abundance of *Akkermansia muciniphila*, which increases the vulnerability of the host to gastrointestinal disorders (*Sovran et al., 2019*). However, the relationships among host gene, intestinal structure and intestinal microbiota have not been fully established.

The intestinal microbiota has effects on muscle growth and development. For example, urease gene-rich microbes, including *Alistipes* and *Veillonella*, help maintain muscle mass in hibernating animals by promoting urea nitrogen salvage (*Regan et al., 2022*), and metabolize lactic acid to provide energy for long periods of exercise and increase endurance in runners (*Scheiman et al., 2019*). Short-chain fatty acids (SCFAs) are gut microbiota-derived metabolites that help maintain the integrity of the intestinal mucosa, improve glucose and lipid metabolism, control energy expenditure, and regulate the immune system and inflammatory responses (*Agus et al., 2021*; *den Besten et al., 2013*). SCFAs are absorbed from the gut lumen and influence host skeletal muscle mass and metabolism (*Lahiri et al., 2019*; *Frampton et al., 2020*). They are involved in the regulation of lipid and glucose metabolism primarily through G-protein-coupled receptors (GPRs), such as GPR41, GPR43, and GPR109 (*Stoddart et al., 2008*; *Van Hul and Cani, 2019*). Although SCFAs, which are gut microbial metabolites, play a role in skeletal muscle development, the mechanism involved requires further clarification.

Slow-twitch muscles are rich in mitochondria and have high oxidative capacity, whereas fast-twitch muscles generate ATP primarily through glycolysis (*Schiaffino and Reggiani, 2011*; *Bassel-Duby and Olson, 2006*). Aging and muscle atrophy result in gradual declines in muscle mass and strength, which are accompanied by an increase in the proportion of type I myofibers, which results in muscle weakness, owing to the preferential loss and atrophy of fast-twitch glycolytic type IIb myofibers (*Akasaki et al., 2014*; *Haber and Weinstein, 1992*; *Faulkner et al., 2007*; *Kirkendall and Garrett, 1998*). Type IIb myofibers are larger in size and more glycolytic, and generate substantial contractile force, but have poorer resistance to fatigue, than type I myofibers (*Schiaffino and Reggiani, 2011*). The activation of the Akt/mTOR pathway was shown to promote the transition of myofibers from the oxidative to the glycolytic myofiber type by increasing expression of the glycolytic proteins hexokinase 2 (HK2), phosphofructokinase-1 (PFK1), and pyruvate kinase isozyme 2 (PKM2) (*Meng et al., 2013*; *Izumiya et al., 2008*; *Verbrugge et al., 2020*).

MSTN affects the growth and function of skeletal muscle. In the present study, we aimed to determine whether the intestinal microbiota remodeled by *MSTN* deletion is involved in the regulation of skeletal muscle growth. Because pigs are similar to humans in many respects, including with respect to their physiology, disease progression, and organ structure (*Swindle et al., 2012*), we used *MSTN*$^{-/-}$ pigs to investigate the regulatory pathway of *MSTN* deletion on intestinal microbiota changes and the

these relationships between intestinal microbiota and skeletal muscle growth and function. Clarify the mechanisms involved in the regulation of muscle growth by the *MSTN* gene-gut microbiota-skeletal muscle axis.

## Results

### *MSTN* deletion stimulates muscle hypertrophy and alters the composition of the gut microbiota in pigs

We used $MSTN^{-/-}$ pigs with 2 and 4 bp deletions in the two alleles of the *MSTN* gene (*Figure 1—figure supplement 1A*) that were generated using the TALEN genome editing technique (*Kang et al., 2017*). We found that $MSTN^{-/-}$ pigs had higher skeletal muscle mass and myofiber cross-sectional area (CSA) but a lack of MSTN expression and lower phosphorylation of Smad2/3 in skeletal muscle (*Figure 1A*–C). In addition, the protein expression of myosin heavy chain (MyHC) type IIb; MyoD; and glycolytic enzymes HK2, PFK1 and PKM2 was significantly higher in skeletal muscle (*Figure 1C and D*).

Because the host genotypes and phenotypes of various mammals have been shown to interact with the gut microbiota (*Kreznar et al., 2017*), we speculated that *MSTN* deletion could affect the composition of the gut microbiota. Therefore, fecal samples from $MSTN^{-/-}$ and wild-type (WT) pigs were compared with respect to the diversity and abundance of gut microbiota using 16 s rRNA-based microbial sequence analysis. The alpha-diversity of the microbial population, which describes its richness and evenness, was evaluated. The ACE of $MSTN^{-/-}$ pigs was significantly lower than that of the WT pigs, but there were no difference in the Chao 1, Shannon, and Simpson indexes (*Figure 1—figure supplement 1B–E*). These results suggest that *MSTN* deficiency causes a reduction in the diversity of the intestinal microbiota. The composition of the gut microbiota, was analyzed using principal components analysis (PCA), which showed that the two groups can be clearly differentiated (*Figure 1E*). LEfSe analysis confirmed a significant difference at the genus level with respect to *Romboutsia* (*Figure 1F*). In addition, *Treponema*, *Romboutsia*, and *Turicibacter*, all of which produce SCFAs (*Kreznar et al., 2017*; *Li et al., 2019b*; *Li et al., 2021*; *Li et al., 2019c*; *Bian et al., 2020*), were significantly more abundant at the genus level in $MSTN^{-/-}$ pigs (*Figure 1G*). Thus, *MSTN* deficiency both stimulates skeletal muscle growth and promotes the growth of microbes that produce SCFAs.

### *MSTN* gene deletion causes changes in intestinal structure and barrier function

The changes in the intestinal environment and barrier function can affect the composition of the gut microbiota (*Dhuppar and Murugaiyan, 2022*; *Bevins and Salzman, 2011*). We observed the muscularis was thicker and the plicae were longer and the expression of the smooth muscle proteins α-SMA and calponin-1 was higher in $MSTN^{-/-}$ pigs (*Figure 2A and B*). In addition, the expression of the tight junction-related genes *TJP1* and *OCLN* was higher (*Figure 2C and D*). Intestinal tight junctions are regulated by MLCK and MLC (*Sinpitaksakul et al., 2008*; *Chun et al., 2014*), and we found that the levels of MLCK and p-MLC were decreased in the intestine of $MSTN^{-/-}$ pigs (*Figure 2D*). IPEC-J2 cells treated with the inhibitor of MSTN receptor SB431542 caused a reduction in MLCK and p-MLC levels, thereby inducing high expression of the tight junction factors *TJP1* and *OCLN* (*Figure 2E and F*). These results demonstrate that *MSTN* gene deletion affects intestinal barrier function through the MLCK/MLC pathway.

### Transplantation of the gut microbiota reshaped by *MSTN* gene deletion promotes the growth of fast-twitch glycolytic muscle

To determine the effects of the intestinal microbiota derived from $MSTN^{-/-}$ pig on skeletal muscle, we transplanted fecal microbes from $MSTN^{-/-}$ and WT pigs into mice. Mice transplanted with WT pig feces were named WT-M, and those transplanted with $MSTN^{-/-}$ pig feces were named KO-M. After 8 weeks of normal chow feeding, the KO-M had higher muscle mass than the WT-M and, in particular, large gastrocnemius (GA) muscles (*Figure 3A*). The GA mass, but not that of the *soleus* (SOL) or *extensor digitorum longus* (EDL), was significantly larger in the KO-M than in the WT-M (*Figure 3B*). However, there were no significant differences in the food intake, physical activity, energy intake, or energy absorbed of the two mouse groups (*Figure 3—figure supplement 1A–E*).

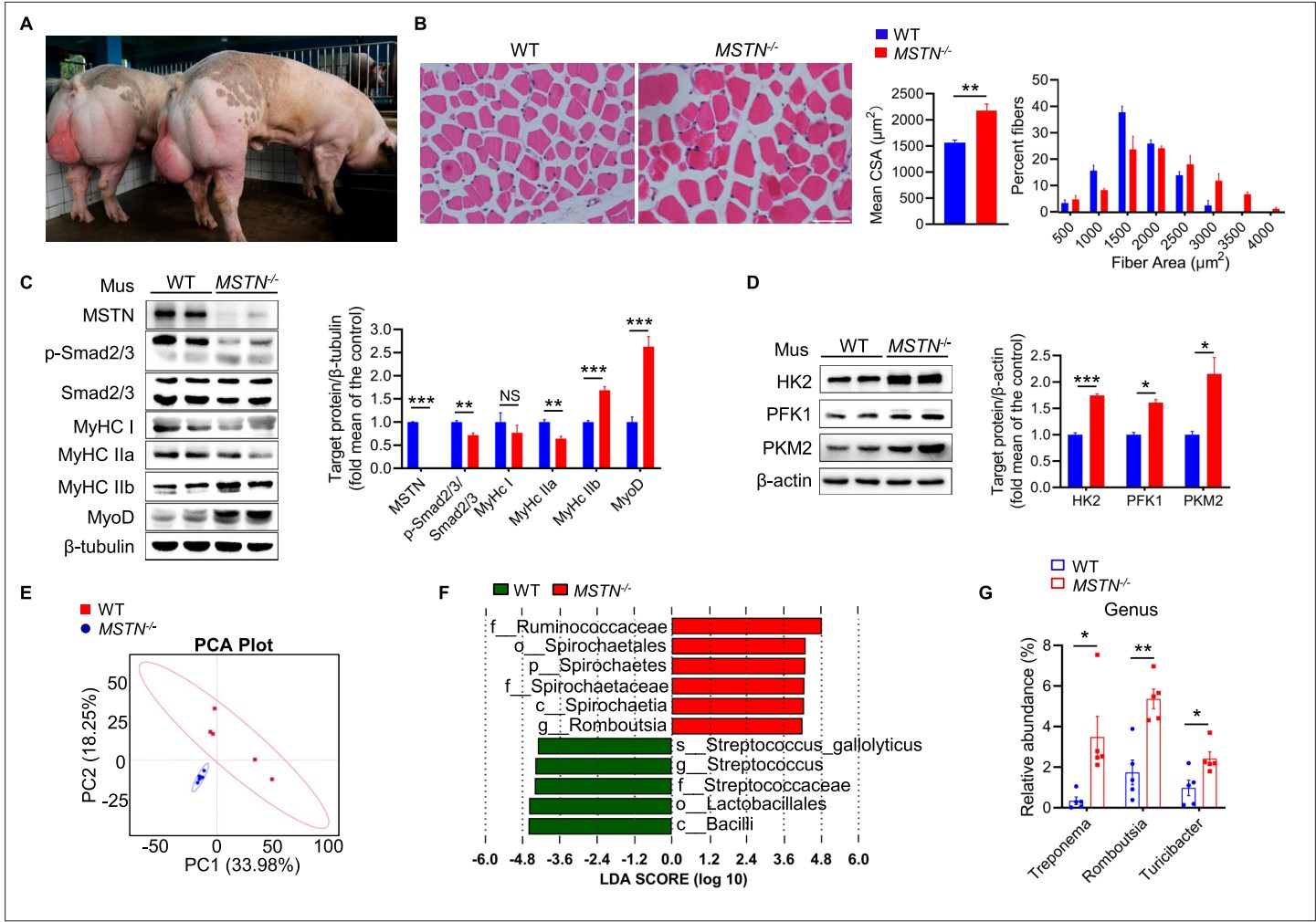

**Figure 1.** *MSTN* deletion stimulates muscle hypertrophy and alters composition of gut microbiota in pigs (n=5). (**A and B**) Representative images of *MSTN*⁻/⁻ pigs and hematoxylin eosin staining *longissimus dorsi*. Magnification is ×200. Scale bar, 100 μm. *MSTN*⁻/⁻ pigs showed skeletal muscle hypertrophy and significantly increased the muscle fiber area. (**C**) Relative to WT pigs, *MSTN*⁻/⁻ pigs showed no expression of MSTN, downregulate phosphorylation of Smad2/3 and MyHC IIa, and upregulate MyHC IIb and MyoD in *longissimus dorsi* (Mus). (**D**) *MSTN*⁻/⁻ pigs showed increased glycolysis enzymes HK2, PFK1 and PKM2 in *longissimus dorsi* (Mus). (**E**) Plots shown were generated using the weighted version of the Unifrac-based PCA. (**F**) Discriminative taxa determined by LEfSe between two groups (log10 LDA >4.8). (**G**) Comparison proportion of genus levels in feces detected by pyrosequencing analysis showed *Treponema*, *Romboutsia*, and *Turicibacter* were increased in *MSTN*⁻/⁻ pigs. Statistical analysis is performed using Student's *t-test* between WT and *MSTN*⁻/⁻ pigs. Data are means ± SEM. *p<0.05; **p<0.01; ***p<0.001; NS, not statistically significant.

The online version of this article includes the following source data and figure supplement(s) for figure 1:

**Source data 1.** Raw data for *Figure 1*.

**Source data 2.** Raw western blot images for *Figure 1C and D*.

**Figure supplement 1.** Gene sequence and alpha-diversity of microbiota of *MSTN*⁻/⁻ pigs.

**Figure supplement 1—source data 1.** Raw data for *Figure 1—figure supplement 1*.

Quantitative analysis of the fiber size of the GA muscles revealed that the fibers were hypertrophic, and that the distribution of fiber size was shifted toward larger fibers in the KO-M (*Figure 3C and D*). As shown in *Figure 3E*, the CSAs of type IIb myofibers of the KO-M were larger than those of the WT-M. Consistent with this, the expression of the proteins MyHC IIb and MyoD, and of the glycolytic enzymes HK2, PFK1, and PKM2, was significantly higher in the GA muscles of the KO-M, whereas the expression of MyHC I and IIa did not significantly differ (*Figure 3F and G*). There was also greater Akt and mTOR phosphorylation in the skeletal muscles of the KO-M (*Figure 3H*). The Akt/mTOR signaling pathway causes type IIb myofiber hypertrophy (*Izumiya et al., 2008*; *Dutchak et al., 2018*), and therefore this may explain the greater GA mass in the KO-M.

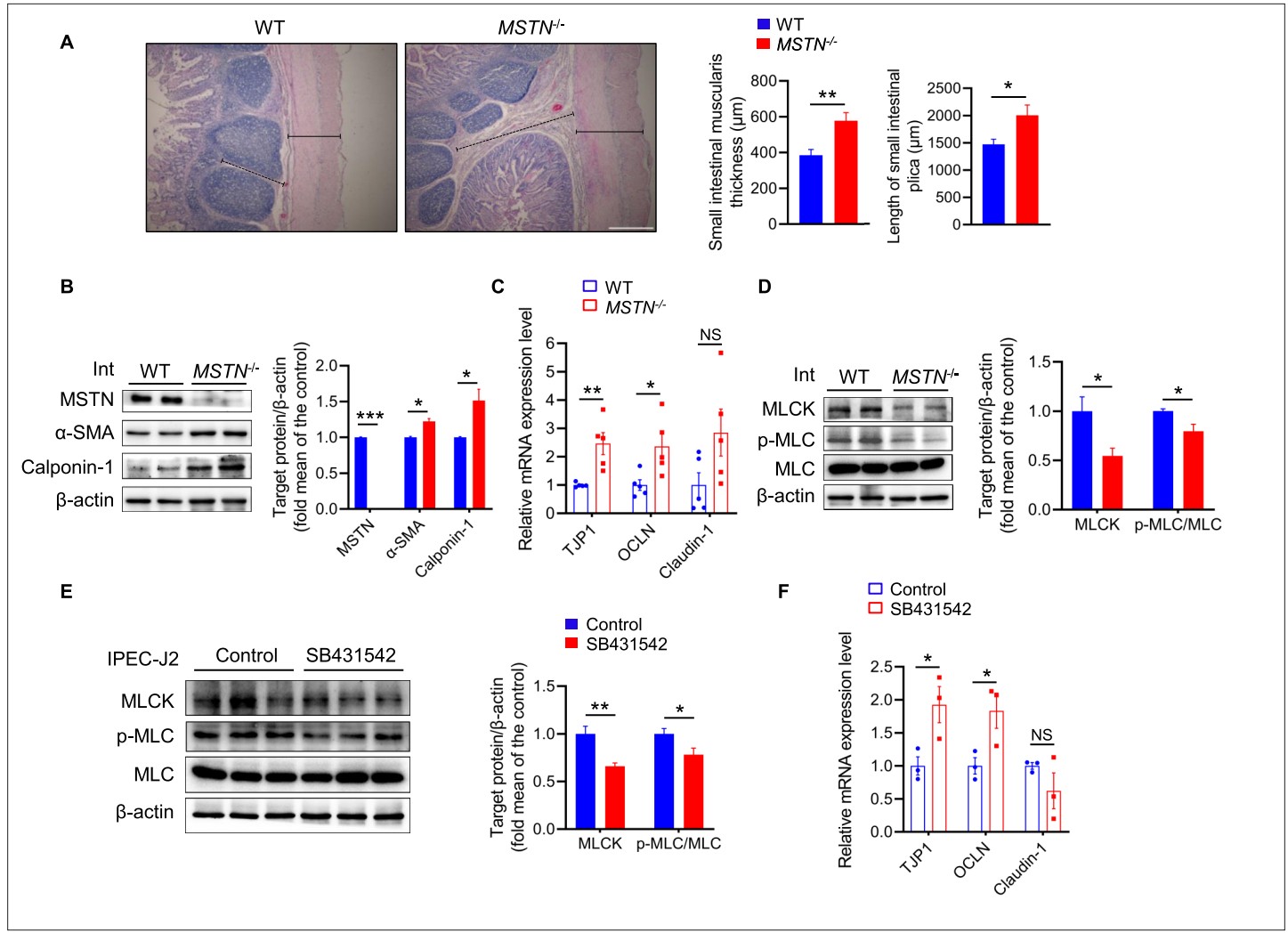

**Figure 2.** *MSTN* deletion alters intestinal structure and tight junction in pigs (n=5). (**A**) Hematoxylin eosin staining of intestinal morphology. The dotted line indicates the length of the plica and the solid line indicates the thickness of muscularis. Magnification is 40×. Scale bar, 500 μm. *MSTN*⁻/⁻ pigs showed an increase of muscularis thickness and plica length in small intestine. (**B**) Relative expression of tight junction genes *TJP1* and *OCLN* were enhanced in small intestine (Int) of *MSTN*⁻/⁻ pigs. (**C**) The protein expression of MSTN was not detected in intestine (Int) while the α-SMA and Calponin-1 were increased in *MSTN*⁻/⁻ pigs compared with the WT pigs. (**D**) The protein expression of MLCK and phosphorylation of MLC in intestine (Int) were decreased in *MSTN*⁻/⁻ pigs compared with the WT pigs. (**E**) The protein expression of MLCK and phosphorylation of MLC were decreased in IPEC-J2 after SB431542 treatment. (**F**) The expression of tight junction factors *TJP1* and *OCLN* were enhanced in IPEC-J2 after SB431542 treatment. Statistical analysis is performed using Student's *t-test*. Data are means ± SEM. *p<0.05; **p<0.01; ***p<0.001; NS, not statistically significant.

The online version of this article includes the following source data for figure 2:

**Source data 1.** Raw data for *Figure 2*.

**Source data 2.** Raw western blot images for *Figure 2B, D and E*.

We also performed a series of physiological experiments to evaluate the strength and running performance of mice that had undergone fecal microbiota transplantation (FMT). Consistent with the expression profile of type IIb myofibers, the grip force of the KO-M was higher than that of the WT-M (*Figure 3I*). However, the KO-M mice had a lower running capacity (*Figure 3J*). These findings may be explained by enlargement in the type IIb myofibers, a type of fast-twitch glycolytic muscle, which are responsible for explosive force rather than endurance. Collectively, these findings imply that the KO-M have larger type IIb myofibers and higher fast-twitch glycolytic skeletal muscle mass.

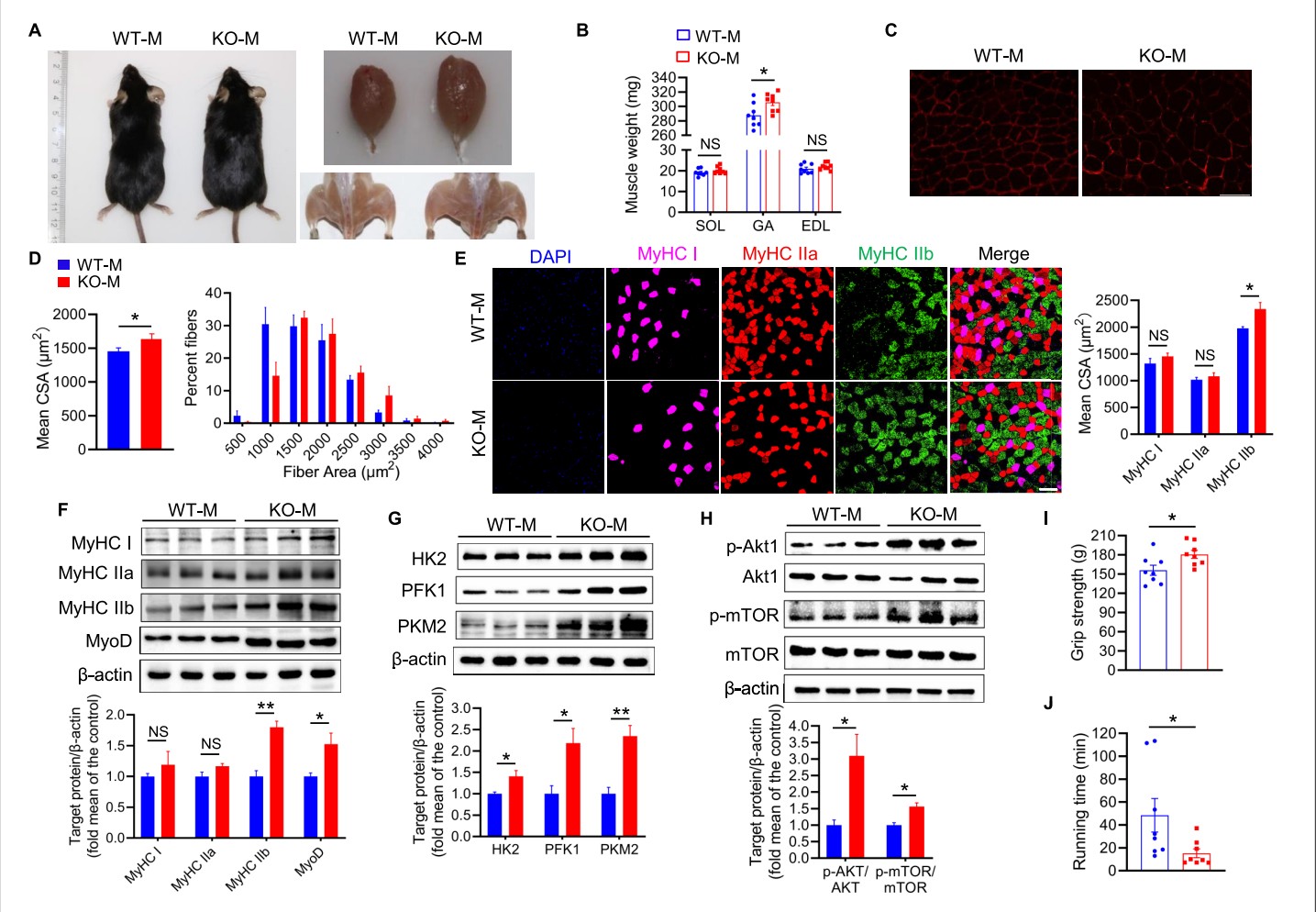

**Figure 3.** Mice fecal microbiota transplantation from *MSTN* deletion pigs induces type IIb myofiber growth. Mice were treated with porcine fecal microbiota for eight weeks by daily oral gavage after combined antibiotics treatment for a week. WT-M, WT pigs fecal microbiota-received mice (n=8); KO-M, *MSTN*⁻/⁻ pigs fecal microbiota-received mice (n=8). (**A**) Representative images of gross appearance and GA of WT-M and KO-M. (**B**) GA mass was increased in KO-M while SOL and EDL were not different between WT-M and KO-M. (**C**) Representative images of GA sections stained with laminin. Magnification is 400×. Scale bar, 50 µm. (**D**) Quantification analysis of myofiber CSA showed that KO-M was larger than WT-M. (**E**) Representative images of GA sections stained with MyHC I (pink), IIa (red), IIb (green) antibodies and nucleuses were stained with DAPI (blue). Magnification is ×200. Scale bar, 100 µm. Quantification of myofiber displayed MyHC IIb CSA were increased in KO-M. (**F**) KO-M showed upregulate the level of MyHC IIb and MyoD in GA. (**G**) The expression of glycolysis enzymes HK2, PFK1, and PKM2 were increased in KO-M GA. (**H**) The Akt/mTOR pathway was activated in KO-M GA. (**I and J**) Grip strength was enhanced while running time was reduced in KO-M compared with WT-M. Statistical analysis is performed using Student's *t-test* between WT-M and KO-M groups. Data are means ± SEM. *p<0.05; **p<0.01; NS, not statistically significant.

The online version of this article includes the following source data and figure supplement(s) for figure 3:

**Source data 1.** Raw data for *Figure 3*.

**Source data 2.** Raw western blot images for *Figure 3F–H*.

**Figure supplement 1.** Food intake, physical activity and energy in mice after fecal microbiota transplantation.

**Figure supplement 1—source data 1.** Raw data for *Figure 3—figure supplement 1*.

## FMT form *MSTN*⁻/⁻ pigs alters the gut microbial composition of the mice

To investigate the relationship between myofiber hypertrophy and the intestinal microbiota of the mice, we analyzed the composition of their intestinal microbiota. There were no significant differences in the ACE, Chao 1, Shannon, or Simpson indexes, indicative of alpha-diversity (*Figure 4—figure supplement 1A−D*). Principal coordinates analysis (PCoA) showed that the composition of the intestinal

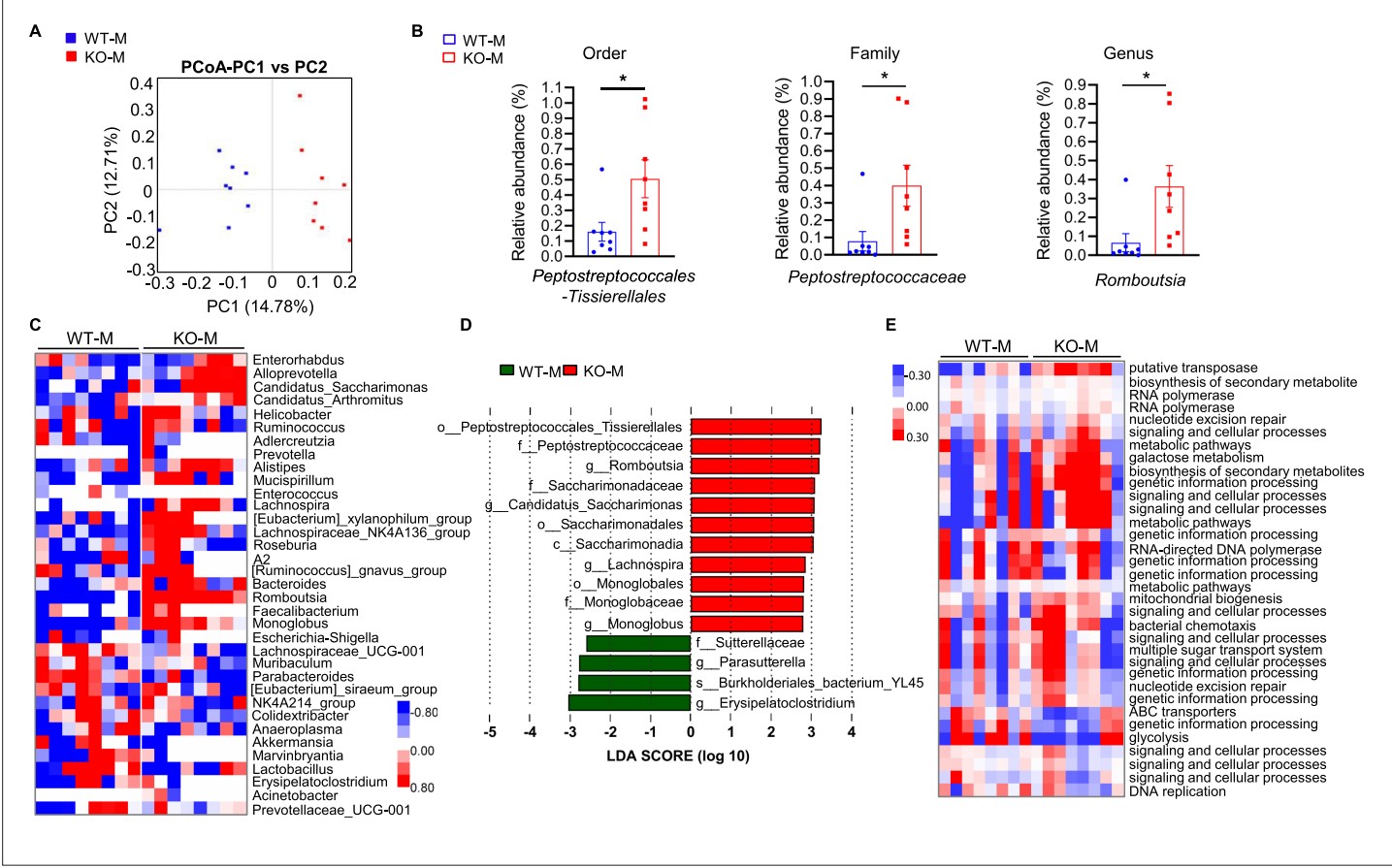

**Figure 4.** *MSTN*<sup>−/−</sup> pigs fecal microbiota transplantation alters microbiota composition in mice. Transplanting fecal microbiota of *MSTN*<sup>−/−</sup> pigs and WT pigs separately to mice (n=8). (**A**) Plots shown were generated using the weighted version of the Unifrac-based PCoA. (**B**) Comparison proportion of order, family and genus levels of *Romboutsia* in feces detected by pyrosequencing analysis. (**C**) Heatmap shows the abundance of top 35 microbial genuses levels was significantly altered by WT and *MSTN*<sup>−/−</sup> donor pigs between WT-M and KO-M groups. (**D**) Discriminative taxa determined by LEfSe between two groups (log10 LDA >3.5). (**E**) Functional prediction shows that intestinal microbial functions are concentrated in functional pathways related to metabolite synthesis after fecal microbiota transplantation. Statistical analysis is performed using Student's *t-test* between WT-M and KO-M groups. Data are means ± SEM. *p<0.05.

The online version of this article includes the following source data and figure supplement(s) for figure 4:

**Source data 1.** Raw data for *Figure 4*.

**Figure supplement 1.** Alpha-diversity in mice after fecal microbiota transplantation from *MSTN*<sup>−/−</sup> pigs.

**Figure supplement 1—source data 1.** Raw data for *Figure 4—figure supplement 1*.

microbiota of the two groups clearly differed (*Figure 4A*). In addition, the genus *Romboutsia* and the corresponding family and order *Peptostreptococcaceae* and *Peptostreptococcales-Tissierellales* were significantly enriched in the intestines of the KO-M (*Figure 4B*). The heat map showed that 22 genera were more abundant, while 13 were less abundant. *Romboutsia*, which was upregulated in *MSTN*<sup>−/−</sup> pigs, was also upregulated in the KO-M (*Figure 4C and D*). Functional prediction analysis showed that intestinal microbial functions of the KO-M were dominated by pathways related to the biosynthesis of secondary metabolites (*Figure 4E*). Thus, mice transplanted with *MSTN*<sup>−/−</sup> pig feces had a larger intestinal population of Romboutsia, and the microbes present are involved in the synthesis of metabolites.

## Gut microbe-derived valeric acid promotes the myogenic differentiation of myoblasts

SCFAs are metabolites of the intestinal microbiota that can affect the growth and function of skeletal muscle (*Frampton et al., 2020*). As described above, FMT of the microbiota of *MSTN*<sup>−/−</sup> pig causes significant increases in skeletal muscle mass and abundance of *Romboutsia*, which can produce

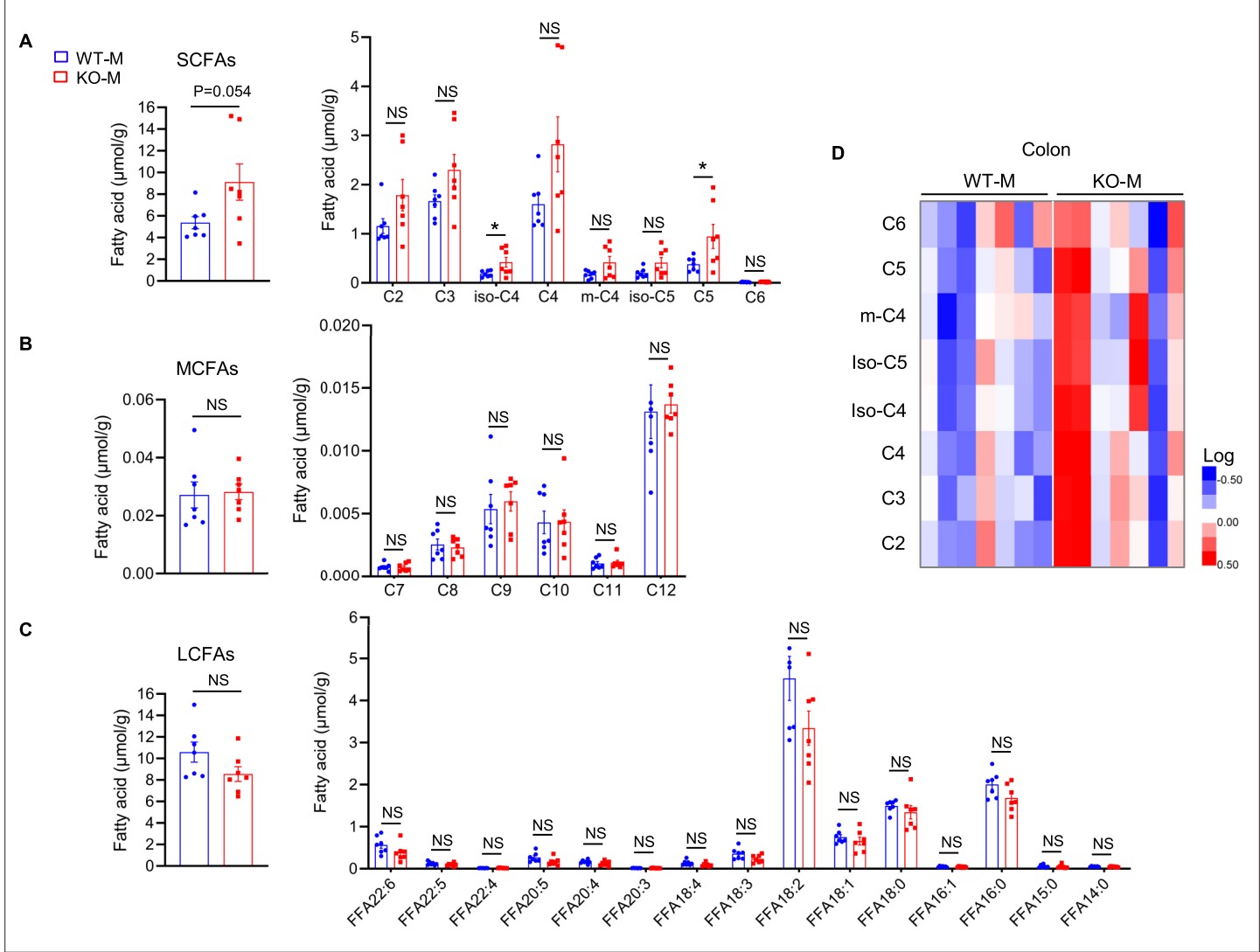

**Figure 5.** *MSTN⁻/⁻* pigs fecal microbiota transplantation alters the level of fatty acids in mice (n=7). (**A**) Fecal microbiota transplantation increased colon total SCFAs (particularly valeric acid and isobutyric acid) in KO-M. (**B**) Fecal microbiota transplantation has no effect on MCFAs between WT-M and KO-M. (**C**) Fecal microbiota transplantation has no effect on LCFAs between WT-M and KO-M. (**D**) Heatmap showed the difference of SCFAs between WT-M and KO-M. Statistical analysis is performed using Student's *t-test*. Data are means ± SEM. *p<0.05; NS, not statistically significant.

The online version of this article includes the following source data for figure 5:

**Source data 1.** Raw data for *Figure 5*.

SCFAs, in recipient mice. Analysis of the fatty acid composition of the colonic contents of the mice showed that SCFAs were present at higher concentrations in the KO-M than in the WT-M. In particular, valeric acid and isobutyric acid were present at significantly higher concentrations in the KO-M (*Figure 5A*). However, there were no differences in the concentrations of medium-chain fatty acids (MCFAs) (*Figure 5B*) or in the concentration of long-chain fatty acids (LCFAs) between the two groups (*Figure 5C*). The heatmap also showed a significant difference in the SCFAs content of the KO-M and WT-M, although there was one outlier in the KO-M group (*Figure 5D*).

To assess the effects of increased concentrations of these SCFAs on myoblast differentiation, the C2C12 myoblast cell line was treated for 24 hr with 5 mM each of valeric acid and isobutyric acid during differentiation. Immunofluorescence staining for MyHC showed that after supplementation with valeric acid, the C2C12 myoblasts produced thicker myotubes and showed a higher fusion index than the control cells, which implies that valeric acid promotes myotube formation (*Figure 6A*). Valeric acid treatment also increased the expression of MyoD and MyoG and promoted the differentiation

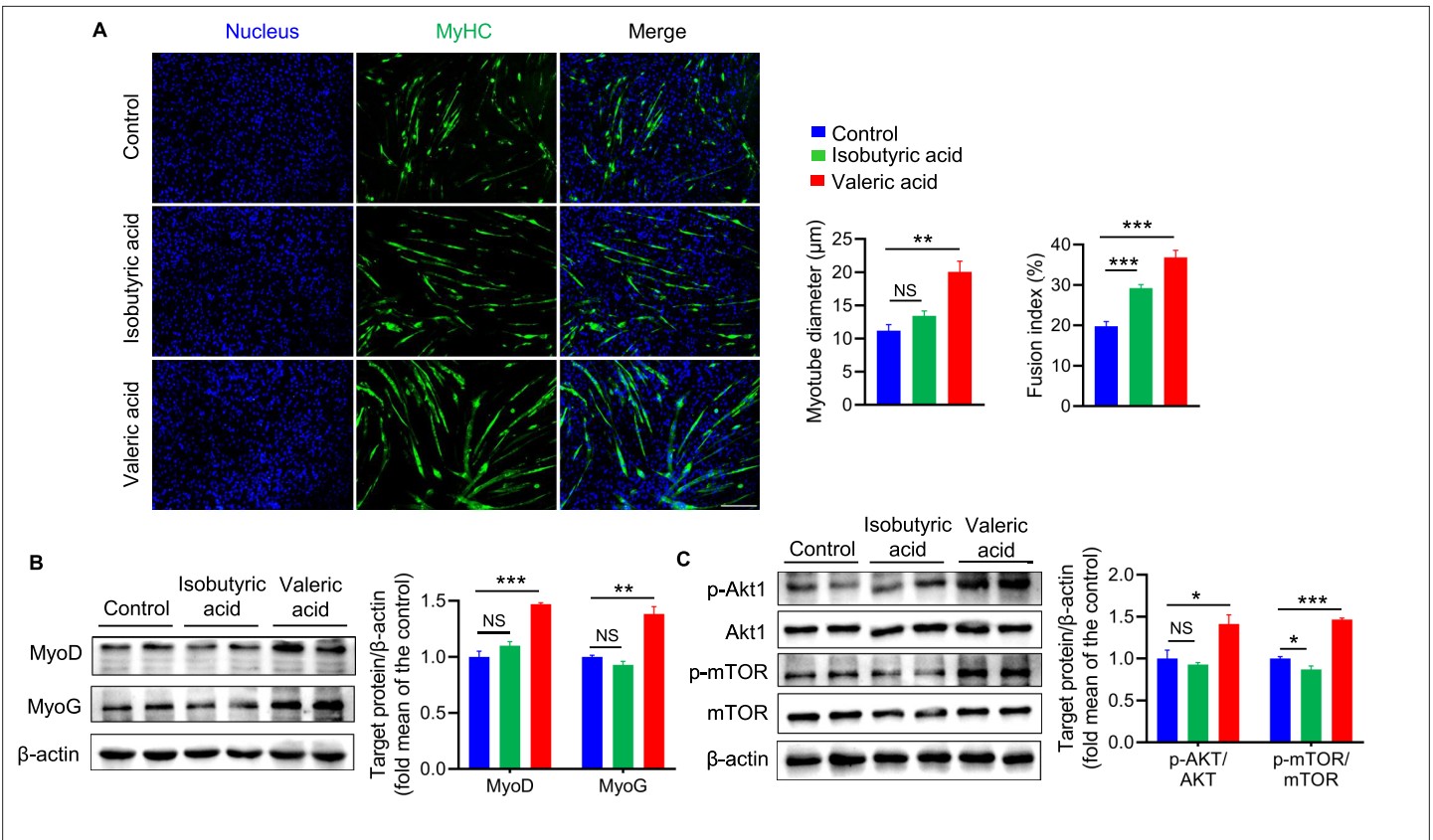

**Figure 6.** Valeric acid treatment promotes myogenic differentiation of myoblast (n=6). (**A**) Representative images of immunofluorescence stained with a specific antibody to identify MyHC (green) of myotubes and the nucleuses were stained with DAPI (blue). Magnification is ×100. Scale bar, 200 μm. Quantification analysis displayed valeric acid treatment increased the diameter and fusion index of myotube, while isobutyric acid only increased the myotube fusion index. (**B**) Valeric acid treatment increased the expression of MyoD and MyoG in C2C12 myoblasts. (**C**) Valeric acid treatment activated the Akt/mTOR pathway. Statistical analysis is performed using one-way ANOVA. Data are means ± SEM. *p<0.05; **p<0.01; ***p<0.001; NS, not statistically significant.

The online version of this article includes the following source data for figure 6:

**Source data 1.** Raw data for *Figure 6*.

**Source data 2.** Raw western blot images for *Figure 6B and C*.

of C2C12 myoblasts (*Figure 6B*). In addition, the phosphorylation of Akt and mTOR was significantly higher following valeric acid treatment (*Figure 6C*). However, isobutyric acid did not have all of these effects, and it only increased the myotube fusion index. Taken together, these results strongly demonstrate that valeric acid can promote myogenic differentiation of myoblasts.

## Valeric acid stimulates the growth of type IIb myofibers

We next evaluated the effect of valeric acid treatment on the skeletal muscle phenotype in vivo. Mice were administered valeric acid (100 mg/kg) daily by oral gavage, which significantly increased the masses of the GA muscles, a fast-twitch glycolytic skeletal muscle, *versus* the control muscles (*Figure 7A and B*). Consistently, valeric acid treatment caused an increase in the CSAs of the GA muscle and in the proportion of large myofibers (*Figure 7C*). In addition, in valeric acid-treated mice, the protein expression of MyHC IIb was significantly higher, that of MyHC I was lower, and that of MyHC IIa showed no difference (*Figure 7D*). Furthermore, valeric acid treatment significantly increased the expression of the glycolytic enzymes HK2, PFK1, and PKM2 (*Figure 7E*); the phosphorylation of Akt and mTOR in the GA muscle (*Figure 7F*); and the grip force of the mice (*Figure 7G*). Interestingly, valeric acid treatment also increased the length of the small intestine (*Figure 7H*) but had no effect on the food intake, physical activity, energy intake, or energy absorbed by the mice (*Figure 7—figure supplement 1A–E*).

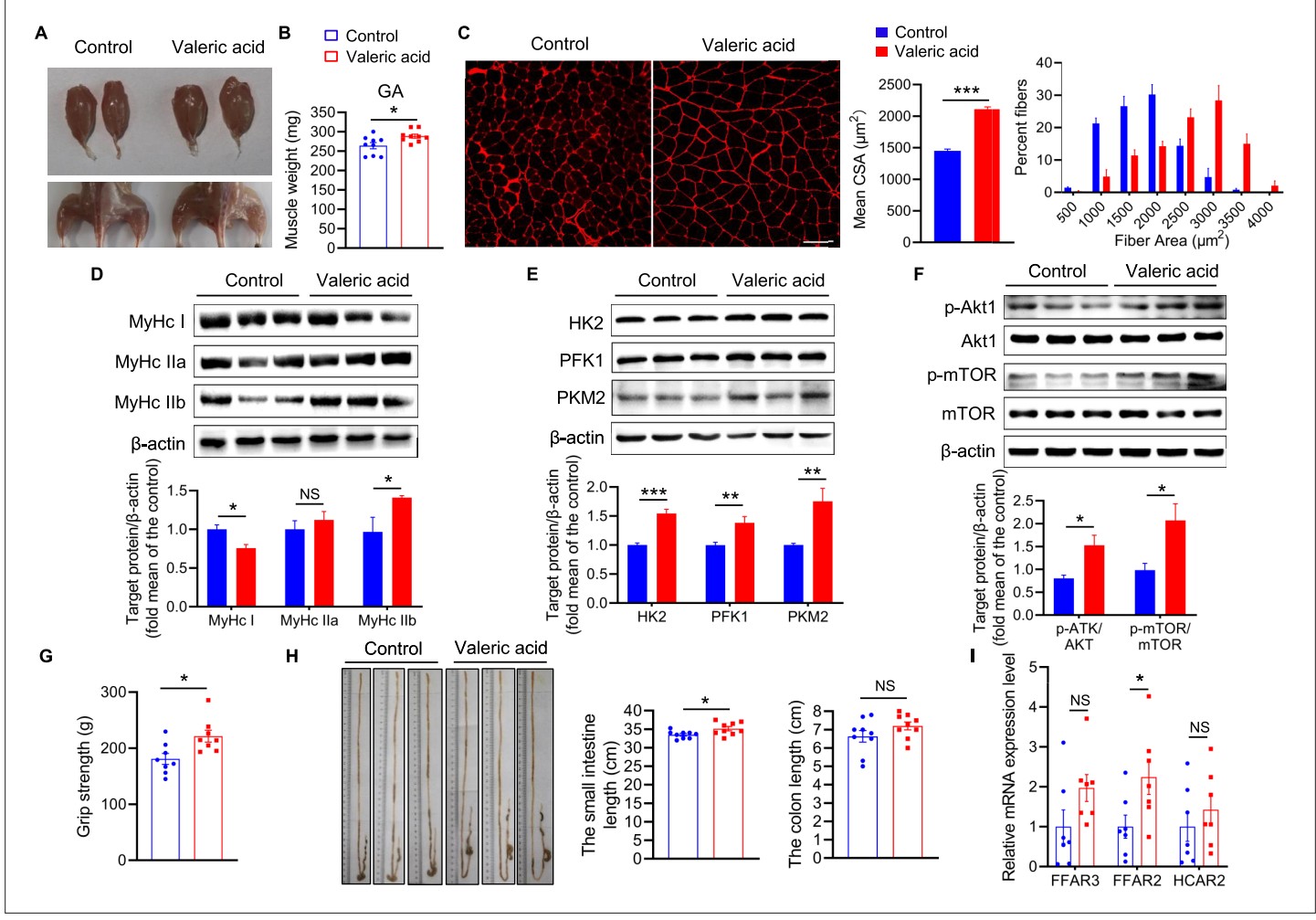

**Figure 7.** Valeric acid induces type IIb myofiber growth and increased GA mass in mice. Mice were treated with valeric acid (100 mg/kg) for 5 weeks by daily oral gavage (n=8–9). (**A**) Representative images of gross appearance and GA of control and valeric acid treated mice. (**B**) Valeric acid treatment increased GA mass. (**C**) Representative images of GA sections stained with laminin, showed valeric acid treatment increased CSA of myofiber. Magnification is ×200. Scale bar, 100 μm. Western blot analysis showed that valeric acid treatment increased the levels of (**D**) MyHC IIb, (**E**) glycolysis enzymes HK2, PFK1, and PKM2, and (**F**) activated the Akt/mTOR pathway in GA compared with control mice. (**G**) Valeric acid treatment improved grip strength. (**H**) Representative images of cecum, small intestine, and colon of mice, showed valeric acid treatment incerased small intestine length. (**I**) Real-time PCR analysis indicated that valeric acid treatment enhanced relative mRNA expression of *FFAR2* in GA. Statistical analysis is performed using Student's *t-test*. Data are means ± SEM. *p<0.05; **p<0.01; ***p<0.001; NS, not statistically significant.

The online version of this article includes the following source data and figure supplement(s) for figure 7:

**Source data 1.** Raw data for *Figure 7*.

**Source data 2.** Raw western blot images for *Figure 7D–F*.

**Figure supplement 1.** Food intake, physical activity and energy in mice after valeric acid treatment.

**Figure supplement 1—source data 1.** Raw data for *Figure 7—figure supplement 1*.

To determine whether the effect of valeric acid on muscle growth is dependent on fatty acid receptors, we first measured the expression of SCFA receptors in skeletal muscle. Valeric acid increased the mRNA expression of *FFAR2* in GA muscle, whereas that of *FFAR3* and *HCAR2* was not affected (*Figure 7I*). Next, mice were orally administered the GPR43-specific inhibitor GLPG0974, which prevented the valeric acid-induced increases in skeletal muscle mass and fiber hypertrophy (*Figure 8A and B*). Consistent with this, the results of *FFAR2* knockdown in C2C12 myoblasts confirmed that valeric acid activates AKT and mTOR via GPR43 (*Figure 8C and D*).

We also compared the effect of valeric acid on myogenic differentiation with that of acetic acid, propionic acid, and butyric acid, and found that acetic acid and propionic acid did not increase the

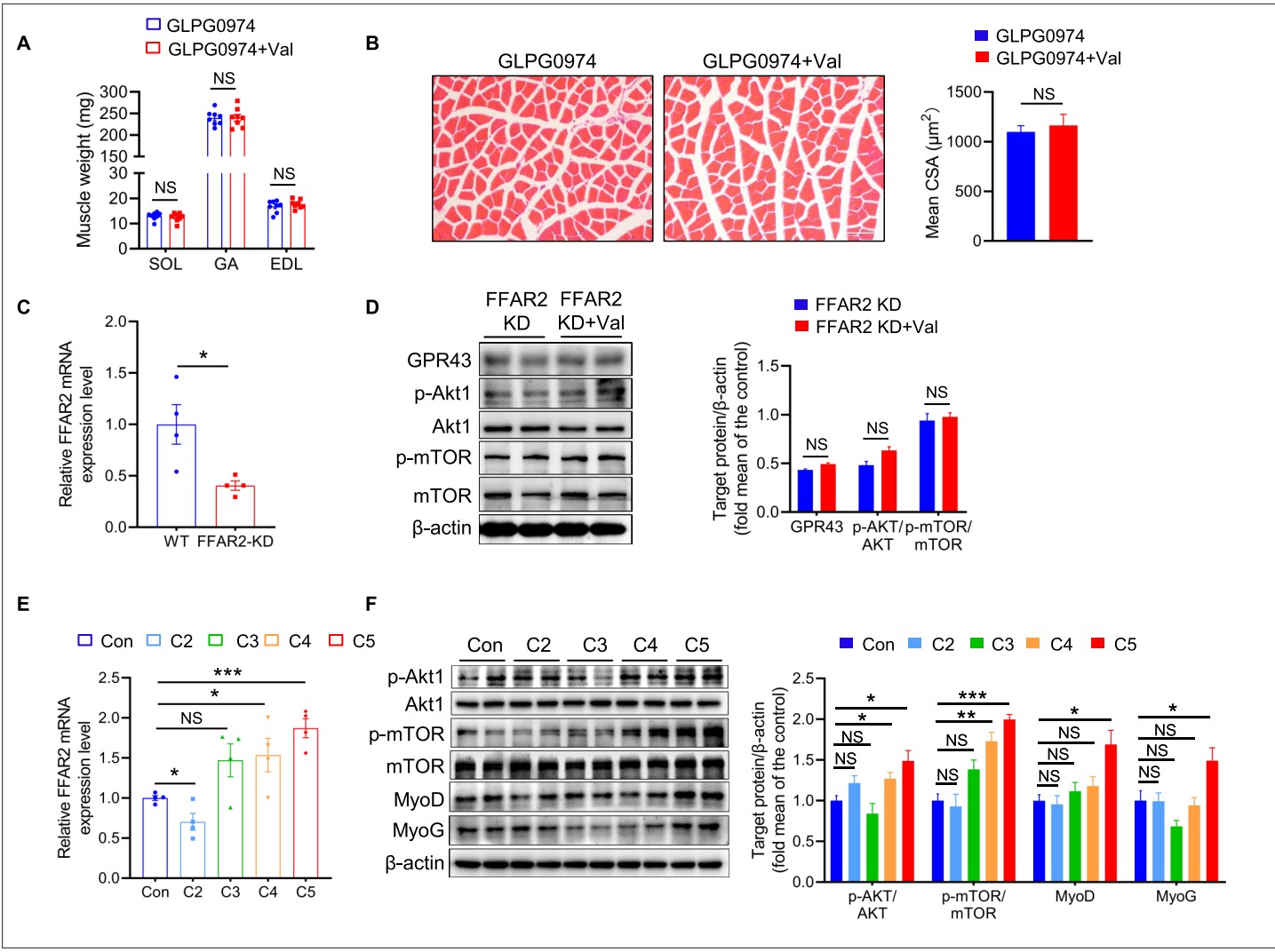

**Figure 8.** Valeric acid promotes skeletal muscle growth by activating the Akt/mTOR pathway through GPR43. (**A**) Mice were treated with valeric acid and GPR43-specific inhibitor GLPG0974 for 5 weeks by oral gavage (n=8), and valeric acid treatment did not increase muscle mass. (**B**) Representative images of GA sections stained with hematoxylin eosin, showed valeric acid treatment did not increase CSA of myofiber when GPR43 was inhibited. Magnification is ×200. Scale bar, 100 μm. (**C**) The mRNA expression of *FFAR2* was knockdown (FFAR2-KD) in C2C12 myoblast. (**D**) Western blot analysis showed that valeric acid treatment did not activate the Akt/mTOR pathway after FFAR2-KD. (**E**) Valeric acid (**C5**) and butyric acid (**C4**) significantly increased the mRNA expression of *FFAR2*, whereas acetic acid (**C2**) and propionic acid (**C3**) did not. (**F**) Valeric acid (**C5**) and butyrate acid (**C4**) activated the Akt and mTOR, and only valeric acid induced high expression of MyoD and MyoG. Statistical analysis is performed using Student's *t-test*. Data are means ± SEM. *p<0.05; **p<0.01; ***p<0.001; NS, not statistically significant.

The online version of this article includes the following source data for figure 8:

**Source data 1.** Raw data for *Figure 8*.

**Source data 2.** Raw western blot images for *Figure 8D and F*.

expression of *FFAR2*, whereas butyric acid and valeric acid significantly increased the mRNA expression of *FFAR2* and the phosphorylation of Akt and mTOR, and only valeric acid induced high expression of MyoD and MyoG (*Figure 8E and F*). These findings suggest that valeric acid induces type IIb/glycolytic myofiber growth and increases GA mass by activating Akt/mTOR signaling *via* GPR43.

## Valeric acid ameliorates dexamethasone (Dex)-induced skeletal muscle atrophy

Glucocorticoids, such as dexamethasone (Dex), are often used experimentally to induce muscle atrophy and are known to influence protein metabolism in skeletal muscle, and high levels are considered

to be a risk factor for the development of muscle atrophy (*Hong et al., 2019*; *Li et al., 2017*). To further explore the effects of valeric acid in skeletal muscle, we used in vivo and in vitro models of Dex-induced muscular atrophy. Valeric acid administration ameliorated the skeletal muscle atrophy induced by Dex in mice and reduced the dissolution area with a clear morphology of muscle fiber (*Figure 9A*). It also significantly reduced the mRNA and protein expression of the pro-atrophic factors atrogin-1 and MuRF-1, the expression of which was induced by Dex (*Figure 9B and C*). In C2C12 myoblasts, valeric acid treatment significantly increased myotube diameter and fusion index, and reduced the expression of the pro-atrophic factors, thereby ameliorating the Dex-induced myotube atrophy (*Figure 9D and E*). Taken together, these findings indicate that valeric acid ameliorates Dex-induced muscle atrophy.

## Discussion

Host genetic variations can influence the composition of gut microbiota, and the gut microbiota can affect skeletal muscle growth and function. In the present study, we found that the gut microbiota was changed by *MSTN* gene deletion in pigs, and that the transplantation of this intestinal microbiota promotes skeletal muscle hypertrophy in mice. Importantly, we have shown for the first time that *MSTN* gene deletion alters the intestinal barrier through the MLCK/p-MLC pathway, which would be expected to affect the composition of intestinal microbiota. Furthermore, we have provided evidence that the intestinal microbiota remodeled by *MSTN* gene deletion increases fast-twitch glycolytic muscle growth through an increase in the production of valeric acid, which activates the Akt/mTOR pathway through the SCFA receptor GPR43. Finally, we have shown that valeric acid has a beneficial effect on the skeletal muscle atrophy induced by Dex (*Figure 10*).

MSTN regulates myogenic differentiation and skeletal muscle mass principally by activating classical Smad2/3 transcription factors (*Chen et al., 2021a*). In the present study, $MSTN^{-/-}$ pigs generated using TALEN genome editing showed inhibition of Smad, enlargement of type IIb myofibers, and overgrowth of skeletal muscle, often referred to as the 'double-muscle' phenotype. These findings are consistent with those of the previous studies performed in *MSTN* mutant mice and cattle (*McPherron et al., 1997*; *Ceccobelli et al., 2022*; *McPherron and Lee, 1997*; *Kambadur et al., 1997*). $MSTN^{-/-}$ pigs exhibit stronger muscle hypertrophy and fiber enlargement, showing a significant 'double-muscle' phenotype. However, the degree of increased muscle mass and fiber size in mice after fecal microbiota transplantation is limited, with only significant increase observed in type IIb muscle fibers. Therefore, the phenotype of increased skeletal muscle mass in mice is not as significant as that in MSTN-KO pigs. We believe that the autonomous changes in muscle cells induced by MSTN KO may be its primary pathway, and our results strongly support the important role of MSTN-mediated changes in gut microbiota in skeletal muscle hypertrophy.

MSTN expression has been identified not only in skeletal muscles but also in the smooth muscle of blood vessels, the penis, and other tissues, where it co-localizes with α-smooth muscle actin and can affect organ function (*Verzola et al., 2017*; *Esposito et al., 2020*; *Kovanecz et al., 2017*). Intestine also contains smooth muscle, and *MSTN* expression in intestine has been demonstrated; however, its role in the intestine is unclear (*Sundaresan et al., 2008*). The present study is the first to show that *MSTN* knockout leads to a loss of expression in the intestine, which was associated with a thicker intestinal muscularis and longer plicae in the pigs, indicating that *MSTN* knockout induces changes in intestinal morphology. The muscularis is important for intestinal motility, and its thickness relates to the peristaltic ability of the intestine. In addition, the height of the mucosal fold determines the surface area available for intestinal absorption (*Wang et al., 2019*; *Zhao et al., 2017*; *Geda et al., 2012*). The identified increases in small intestinal muscularis thickness and plical length imply that $MSTN^{-/-}$ pigs have greater intestinal absorptive capacity.

Tight junctions between adjacent intestinal epithelial cells are a critical component of the intestinal barrier (*Ghosh et al., 2020*). Previous studies have shown that a disruption in the intestinal barrier leads to greater entry of bacterial products, including lipopolysaccharide, into the circulation, which triggers an inflammatory response in specific tissues, such as skeletal muscle and adipose tissue (*Ghosh et al., 2020*). In the present study, the expression of the tight junction factor *TJP1* and *OCLN* was significantly higher in the intestines of $MSTN^{-/-}$ pigs. Intestinal barrier function is regulated by the MLCK/MLC pathway (*Sinpitaksakul et al., 2008*; *Chun et al., 2014*), and interestingly, the expression of MLCK and the phosphorylation of MLC were significantly lower in $MSTN^{-/-}$ pigs. These results

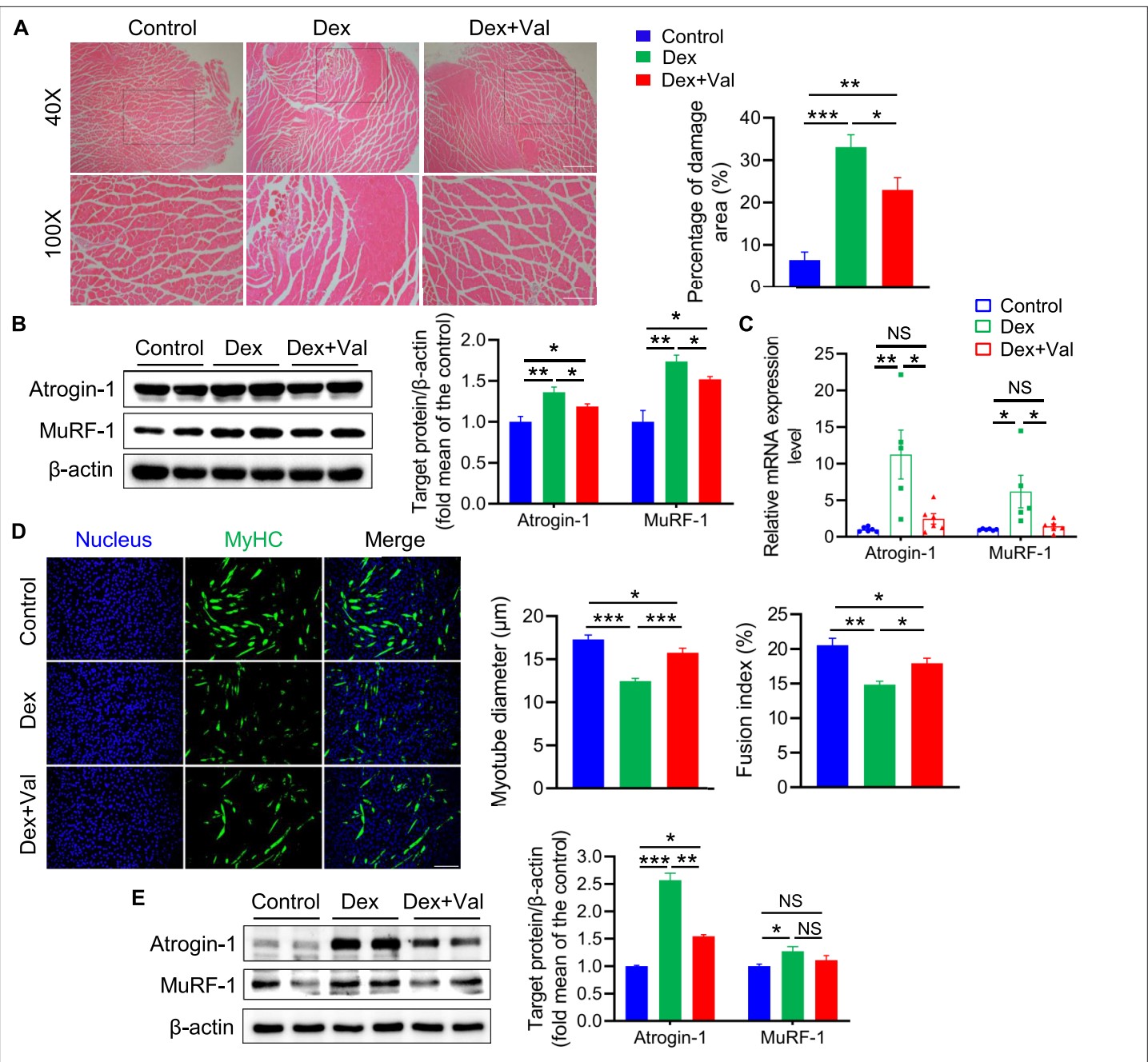

**Figure 9.** Valeric acid ameliorates Dex-induced skeletal muscle and myotube atrophy. Mice were treated with intraperitoneal injection of 20 mg/kg Dex every 2 day for 2 weeks and 100 mg/kg of valeric acid was fed orally every day before 2 weeks of Dex injection (n=5). Myotube atrophy was induced with 100 μM/L of Dex, and 5 mM/L of valeric acid was supplied at the same time (n=6). (**A**) Hematoxylin eosin staining of GA morphology. Magnification is ×40. Scale bar, 500 μm. Quantification analysis showed that Dex induced the myofiber damage, and valeric acid treatment decreased the percentage of damage area. (**B**) Western blot analysis showed that Dex induced the expression of Atrogin-1 and MuRF-1 in GA, while valeric acid treatment reduced the level of these. (**C**) Real-time PCR analysis of relative expression of atrophy genes (*Atrogin-1* and *MuRF-1*) in GA, showed valeric acid treatment could inhibit the expression of these genes induced by Dex. (**D**) Immunofluorescence stained with a specific antibody was used to identify MyHC (green) of myotube and the nucleus were stained with DAPI (blue). Magnification is ×100. Scale bar, 200 μm. Quantification analysis showed valeric acid treatment could improve the reduction of myotubes diameter and fusion index induced by Dex. (**E**) Western blot analysis showed valeric acid treatment could inhibit the expression of Atrogin-1 induced by Dex in C2C12 myotubes and had no effect on MuRF-1 induced by Dex. Statistical analysis is performed using one-way ANOVA with *Least Significant Difference test*. Data are expressed as means ± SEM. *p<0.05; **p<0.01; ***p<0.001; NS, not statistically significant.

The online version of this article includes the following source data for figure 9:

*Figure 9 continued on next page*

*Figure 9 continued*

**Source data 1.** Raw data for *Figure 9A*.

**Source data 2.** Raw data for *Figure 9B–E*.

**Source data 3.** Raw western blot images for *Figure 9B and E*.

suggest that MSTN may regulate intestinal tight junction function in pigs through the MLCK/MLC pathway. The IPEC-J2 cell line has morphological and functional similarities to porcine intestinal cells, and it therefore represents a good model for the assessment of intestinal barrier function (*Brosnahan and Brown, 2012*). We found that inhibition of the MSTN receptor caused down regulation of MLCK and p-MLC and improved the tight junctions of IPEC-J2 cells. Therefore, we believe that the deletion of the *MSTN* gene in the intestine can affect the intestinal environment through the MLCK/MLC pathway. Importantly, changes in the intestinal environment and barrier function can alter the microbial composition of the gut (*Seganfredo et al., 2017*; *Nicoletti et al., 2017*; *Sekirov et al., 2010*; *Tremaroli et al., 2015*). The composition of the intestinal microflora of *MSTN*⁻/⁻ pigs was analyzed, and we found that the genera *Romboutsia*, *Treponema*, and *Turicibacter* were significantly more abundant. The results of several previous studies have suggested that these microbes produce SCFAs (*Li et al., 2019b*; *Li et al., 2021*; *Li et al., 2019c*; *Bian et al., 2020*). In addition, *Romboutsia* (*Li et al., 2021*; *Yanni et al., 2020*) and *Turicibacter* (*Watanabe et al., 2021*) are closely associated with metabolic disorders, such as hypertension, diabetes, the dysregulation of skeletal muscle energy metabolism, and obesity. Therefore, the alterations in intestinal structure and barrier function induced by the deletion of the *MSTN* gene in the intestine may affect the composition of the intestinal microbiota.

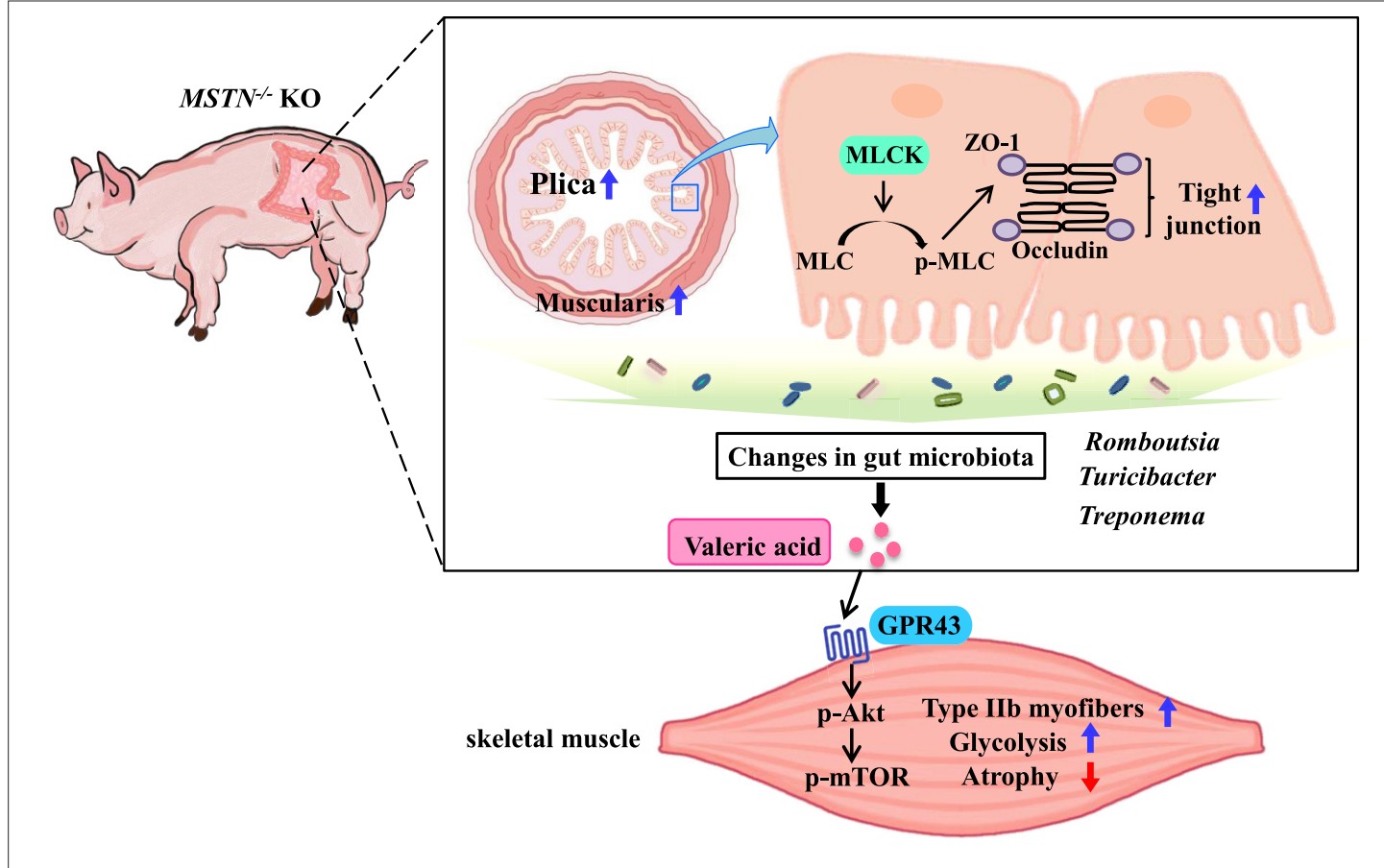

**Figure 10.** Schematic illustration of the results. Intestine MSTN deficiency inhibited MLCK/MLC, altered the intestinal structure and barrier, and reshaped gut microbiota; gut microbiota metabolite-valeric acid activates Akt/mTOR pathway *via* GPR43 to stimulate fast-twitch glycolytic skeletal muscle growth.

FMT can lead to the transfer of both the host gut characteristics and metabolic phenotype from pigs to mice (*Yang et al., 2018*; *Diao et al., 2016*; *Yan et al., 2016*). To determine the effect of the intestinal microbiota remodeled by *MSTN* gene deletion on skeletal muscle, the intestinal microbiota of *MSTN*$^{-/-}$ and WT pigs were transplanted into mice. Interestingly, we found that mice transplanted with *MSTN*$^{-/-}$ pig feces had larger GA muscles and muscle fibers (*Figure 3*), and in particular larger type IIb myofibers, implying that the increase in size of the GA may be attributable to hypertrophy of type IIb myofibers. Previous studies have shown that *Akt1* transgene activation specifically increases GA mass and type IIb myofiber size through an mTOR-dependent pathway (*Izumiya et al., 2008*). This is consistent with the findings of the present study, in which significant activation of the Akt/mTOR pathway was identified alongside increases in GA mass and IIb myofiber CSA in the KO-M. In addition, augment of type IIb myofiber was associated with an increase in grip strength but a reduction in endurance during treadmill testing (*Izumiya et al., 2008*). In the present study, as expected, the KO-M had significantly higher grip strength but poor resistance to fatigue. Importantly, we found that SCFAs producing microbes *Romboutsia*, which was enriched in the *MSTN*$^{-/-}$ pigs, was also significantly enriched in the recipient mice (*Figure 4B*–D). The concentration of SCFAs was significantly higher in the colonic contents of the KO-M mice. Moreover, previous research has shown that treatment with the microbial metabolite SCFAs also ameliorates impairments in skeletal muscle, and especially in the GA muscle, by increasing MyoD expression and reducing atrogin-1 expression (*Lahiri et al., 2019*). In C2C12 cells, SCFAs also stimulates mitochondrial respiration and promotes a switch of energy production mode from oxidative phosphorylation to glycolytic (*Lahiri et al., 2019*). These results strongly suggest that the intestinal microbiota associated with *MSTN* gene deletion causes a growth of fast-twitch glycolytic muscle and improves function, which may be mediated by the enrichment of SCFA-producing microbes.

SCFAs are the principal metabolites generated by the intestinal microbiota and are involved in multiple physiological processes in the host (*Donohoe et al., 2011*; *Canfora et al., 2015*). We observed that valeric acid treatment increases myotube formation in myoblasts, and increases the mass of GA muscles and type IIb muscle fiber size in mice; meanwhile, in previous studies, acetic acid has been shown to improve mitochondrial metabolism and promote the formation of oxidative muscle fibers (*Pan et al., 2015*). In addition, valeric acid treatment increases the length of the small intestine, thereby improving its ability to absorb nutrients. This is conducive to muscle growth and development. SCFAs play a regulatory role principally by binding to GPR41, GPR43, and GPR109a (*Stoddart et al., 2008*; *Van Hul and Cani, 2019*). In the present study, we found that *FFAR2* expression was higher in valeric acid-treated C2C12 cells and mice. Furthermore, the inhibition of GPR43 in mice and the knock down of *FFAR2* in myoblasts were found to prevent valeric acid-induced Akt/ mTOR pathway activation, which implies that valeric acid promotes skeletal muscle growth by activating the Akt/mTOR pathway *via* the SCFA receptor GPR43. Because other SCFAs, such as acetic acid (*Pan et al., 2015*) and butyric acid (*Bian et al., 2020*), also promote skeletal muscle development, we analyzed the effects of various SCFAs in myoblast differentiation, and found that both valeric acid and butyric acid activate the Akt/mTOR pathway via GPR43, whereas acetic acid and propionic acid do not (*Figure 8*). In addition, only valeric acid was found to increase the expression of the myogenic differentiation factors MyoD and MyoG. Aging and long-term or high-dose glucocorticoid therapy induce skeletal muscle atrophy, which principally manifests as a loss of skeletal muscle mass, with a relatively selective loss of type IIb muscle fibers (*Akasaki et al., 2014*; *Haber and Weinstein, 1992*; *Faulkner et al., 2007*; *Kirkendall and Garrett, 1998*). We found that valeric acid treatment ameliorates Dex-induced myotube atrophy and partially repairs skeletal muscle atrophy (*Figure 9*).

In conclusion, this is the first study to demonstrate that *MSTN* gene deletion in pig alters intestinal structure and function, leading to changes in the composition of the intestinal microbiota. We have further demonstrated that *MSTN* gene deletion-induced remodeling of the intestinal microbiota results in the selective hypertrophy of fast-twitch glycolytic muscles in recipient mice. Finally, we have shown that the microbial metabolite valeric acid promotes myoblast differentiation and fast-twitch glycolytic myofiber growth by activating the Akt/mTOR pathway *via* the SCFA receptor GPR43, and ameliorates skeletal muscle atrophy induced by Dex. These findings increase our understanding of the effect of host genetic variation on the gut microbiota, and provide insights into potential new treatments for muscle diseases such as muscular dystrophy and sarcopenia.

## Materials and methods

### Animals

The study was approved by the Ethics Committee of Yanbian University (approval number SYXK2020-0009). We generated $MSTN^{-/-}$ pigs with 2 and 4 bp deletions in the two alleles of the *MSTN* gene using the TALEN genome editing technique and somatic cell nuclear transfer (*Kang et al., 2017*). The pigs were fed a standard commercial diet and housed in the same environmentally controlled room on a pig-breeding farm. Male C57BL/6 mice aged 4 weeks were purchased from Vital River Laboratory Animal Technology (Beijing, China), and chow diet (Beijing HuaFuKang Bioscience, Beijing, China) and water were provided ad libitum. The mice were administered valeric acid (100 mg/kg, Shanghai Aladdin, China) or water (vehicle) by oral gavage from 4 weeks of age; after 5 weeks of treatment, they were euthanized, and their tissues were collected. The GPR43 antagonist GLPG0974 (10 mg/kg) was administered orally every 2 days for 5 weeks, during which time valeric acid was administered daily.

To establish a model of Dex-induced muscle atrophy, male C57BL/6 mice aged 8 weeks were intraperitoneally injected with 20 mg/kg Dex or saline every other day for 2 weeks. Dex-induced skeletal muscle atrophy was confirmed by weight loss in the mice (*Hong et al., 2019*; *Li et al., 2017*). A total of 100 mg/kg valeric acid was administered orally to the mice every day for 2 weeks before Dex injection and until the end of the experiment. The mice were housed in a specific pathogen-free environment at 21±1 °C and 40–60% relative humidity, under a 12/12 h light/dark cycle. For all experiments, animals were fasted overnight before they were euthanized.

### Wheel running and grip strength

Before the wheel running experiment, all mice were trained to run aton at a low speed for a week. Mice were individually housed in running wheels (SA102, Jiangsu SANS Bioscience, China), according to the manufacturer's recommendations. Briefly, the mice ran on the wheel at a speed of 30 r/min until they dropped, and the time and distance they ran were recorded. The mice were trained for a week to use their limbs to grasp a grip meter (SA417, Jiangsu SANS Bioscience, China) before the grip strength test. Each mouse was tested for the highest peak strength by making them pull the grip dynamometer horizontally.

### Physical activity and energy metabolic

Each mouse was individually measured by a small animal activity detector according to the manufacturer's instructions (SA-YLS-1C, Jiangsu SANS Bioscience, China). The physical activity of mice was measured in a quiet environment for 48 hr. Food intake and fecal output were collected and weighed, and food loss was subtracted.

The collected mouse feces were dried at 65 °C for 24 hr, and the calorific value in dry feces and food was accurately measured in isothermal 22 °C by oxygen bomb calorimeter (IKAc2000 basic, Germany). Each mouse's daily energy intake, energy absorbed, energy expelled and energy unabsorbed were calculated by caloric.

### Fecal microbial transplantation

Fecal samples were collected daily from 6-month-old $MSTN^{-/-}$ and WT donor pigs in the morning. In a sterile environment, they were homogenized and suspended in sterile saline (250 mg/mL), and the mixture was centrifuged at 800×*g* for 5 min. Antibiotic mixture (50 µg/mL streptomycin, 100 U/mL penicillin, 170 µg/mL gentamycin, 100 µg/mL metronidazole, and 125 µg/mL ciprofloxacin; Sigma-Aldrich, St. Louis, MO, USA) was added to sterile drinking water and provided daily for 1 week prior to FMT. From 5 weeks of age, each group of recipient mice were gavaged with 200 µL of the corresponding bacterial suspension daily for 8 weeks until euthanasia and tissue collection.

### Analysis of the gut microbiota

Fecal samples were collected for microbial analysis from donor pigs when they were 6-month-old and from the recipient mice after 8 weeks of FMT. The methods used to analyze the diversity and taxonomic profiles of the gut microbiota of the donor pigs and recipient mice have been described previously (*Quan et al., 2020*). Briefly, the CTAB method was used extract the genomic DNA from

the fecal bacteria, and then DNA samples with final concentrations of 1 ng/µL were subjected to bacterial 16 s rRNA gene amplification sequencing (V3–V4 regions). The Illumina NovaSeq platform (Novogene, Beijing, China) was used to determine the abundance and diversity of intestinal microbial taxa in the pigs and mice. The library quality was assessed using a Qubit@ 2.0 Fluorometer (Thermo Scientific, USA) and Agilent Bioanalyzer 2100 system.

Paired-end reads were allocated according to the unique barcodes of the sample and truncated by removing the barcode and primer sequences. FLASH (v1.2.7) was used to merge the overlapping reads between paired-end reads and then according to the QIIME (V1.9.1) quality control process, high-quality clean tags were obtained by qualitative filtration of the original reads under specific conditions. The effective tags were collected after comparison with sample tags in the reference database (Silva database) after the identification and removal of chimeric sequences using the UCHIME algorithm. The QIIME software was used to calculate all indices for the samples, and R (v2.15.3) was used for bioinformatic analyses of the sequences. The equivalent operational taxonomic units had at least 97% sequence similarity. The alpha diversity and beta diversity of the samples were assessed, and PCA was performed according to the unweighted unifrac distances.

## Cell culture

IPEC-J2 and C2C12 myoblasts were procured from the BeNa Culture Collection (Beijing, China) and the National Laboratory Cell Resource Sharing Service Platform (Beijing, China), respectively. The identity was not authenticated by our hands. Cells were free from mycoplasma contamination confirmed by tests for mycoplasma. They were cultured in Dulbecco's modified Eagle's medium (DMEM; Invitrogen-Gibco), containing 10% fetal bovine serum (Sigma), 100 U/mL penicillin and 100 U/mL streptomycin (Invitrogen-Gibco). IPEC-J2 were grown on 6-well plates and treated with SB431542 (MCE, China) for 24 hr after the formation of a confluent monolayer.

For myoblasts differentiation, C2C12 were grown on six-well plates until 80% confluence and then induced to differentiate in DMEM containing 2% horse serum (Invitrogen). SCFAs (valeric acid, acetic acid, propionic acid, butyric acid, or isobutyric acid) were added to the differentiation medium for 24 hr (*Maruta and Yamashita, 2020*; *Han et al., 2014*; *Tang et al., 2022*). The cells were supplied with fresh differentiation medium every 2 days. Myotubes were obtained after 5 days of differentiation. C2C12 myoblasts at 80% confluence were transfected with 1 mg/mL of CRISPR/Cas9 compound plasmid (FFAR2_eSpCas9-2A-GFP) with a sgRNA sequence of AAGATCGTGTGCGCGCTGAC.

To establish the model of Dex-induced myotube atrophy, myoblasts were treated with 100 µmol/L of Dex at the beginning of differentiation for 24 hr, and 5 mmol/L valeric acid was added to some of the cultures. The myoblasts were cultured in fresh differentiation medium for 5 days and the myotubes obtained were immunostained using anti-MyHC antibody (MyHC, A4.1025, Sigma), and Alexa Fluor 488-labelled goat anti-mouse IgG as the secondary antibody (Jackson ImmunoResearch Laboratories). The nuclei were counterstained using 10 µg/µL DAPI (D-9106, Beijing Bioss Biotechnology). The diameters and numbers of nuclei in the differentiated myotubes were measured using Image J (1.51q, National Institutes of Health, Bethesda, MD, USA). For each treatment, five pictures were obtained from each well of the six-well plates. The diameters of three different parts of each myotube were measured, and the mean values were calculated. To determine the C2C12 fusion index, the numbers of nuclei in the myotubes were counted, divided by the total numbers of nuclei, and multiplied by 100.

## Histological analysis

Skeletal muscle and intestinal morphology were examined following hematoxylin and eosin (HE) staining. Five-µm-thick paraffin-embedded sections of the *longissimus dorsi* muscles and intestines from *MSTN*$^{-/-}$ and WT pigs were prepared, and images of the stained sections were obtained using a light microscope (BX53, Olympus, Japan).

Liquid nitrogen-cooled isopentane was used to rapidly freeze the skeletal muscle samples, which were embedded in OCT compound (Sakura Finetech USA Inc). Cryostat sections (10 µm) were prepared from the midbelly of the muscles, and the sections were immunostained using MyHC type I (BA-D5, DSHB, Douglas Houston), MyHC type IIa (SC-71, DSHB, Douglas Houston), MyHC type IIb (BF-F3, DSHB, Douglas Houston), and laminin (ab11575, Abcam) monoclonal antibodies for fiber typing. Alexa Fluor 647-conjugated goat anti-mouse IgG2b, Alexa Fluor 488-conjugated goat anti-mouse IgG1, Alexa Fluor 555-conjugated goat anti-mouse IgM, or Alexa Fluor 594-conjugated goat

anti-rabbit IgG were used as the secondary antibodies. The nuclei were counterstained using 10 μg/μL of DAPI. Fluorescence was detected using a confocal laser scanning microscope (FV3000, Olympus, Tokyo, Japan). Image J software was used to measure the thickness and the CSA of the myofibers. The areas of damage to the skeletal muscle fibers were evaluated by calculating the ratio of the muscle fiber ablation area to the total muscle fiber CSA.

## Quantitative real-time PCR

Total RNA was extracted from liquid nitrogen quick-frozen tissue using a Total RNA Extraction Kit (LS1040; Promega) as per the manufacturer's protocol. After evaluating the concentration and purity of RNA, an equal amount of RNA was used for reverse transcription. Information regarding the primers used is available in the *Supplementary file 1*. Real-time PCR was performed using a Mx3005P system (Agilent, Santa Clara, CA, USA), and the relative gene expression levels were calculated using the $2^{-CT}$ method and normalized to those of the control group.

## Western blotting

The cells and tissues were homogenized in RIPA buffer (Beyotime). The protein concentrations of the lysates were measured using a BCA kit (Beyotime, Shanghai, China), and then immunoblot analysis was performed according to standard procedures. Samples containing equal amounts of protein were electrophoresed and transferred to membranes, which were blocked and incubated with the following primary antibodies: phospho-Akt (Ser473), phospho-mTOR (Ser2448), phospho-Smad2 (Ser465/467)/ Smad3 (Ser423/425), Akt, mTOR, Smad2/3, and HK2 (Cell Signaling Technology), PFK1 and PKM2 (Shanghai Absin, Inc), MyHC type I, MyHC type IIa, and MyHC type IIb (DSHB), MSTN, MyoD, MyoG, α-SMA, calponin-1, MuRF-1, atrogin-1, GPR43, MLCK, p-MLC, MLC, β-actin, and tubulin (Beijing Bioss Biotechnology, Inc). The ChemiDoc MP Imaging System and Image Lab software (Bio-Rad, Hercules, CA, USA) were to analyze the specific bands obtained.

## SCFAs analysis

Colon contents were collected after 8 weeks of FMT. The SCFAs were extracted from the mouse feces using 1:1 acetonitrile/water, derivatized using 3-nitrophenylhdyrazones, and analyzed using a Jasper HPLC coupled to a Sciex 4500 MD system (LipidALL Technologies Co., Ltd, Changzhou, China). Briefly, a Phenomenex Kinetex $C_{18}$ column (100×2.1 mm, 2.6 μm) was used to separate the individual SCFAs, with a mobile phase A consisting of 0.1% formic acid aqueous solution and a mobile phase B consisting 0.1% formic acid/acetonitrile. Octanoic acid-1-$^{13}C_1$ (Sigma-Aldrich) and butyric-2,2-$d_2$ (CDN Isotopes) were used as internal standards for the quantification (*Li et al., 2019a*).

## Free fatty acids (FFAs) analysis

FFAs were extracted from mouse feces using a modified version of the Bligh and Dyer's method (LipidALL Technologies Co., Ltd, Changzhou, China). Briefly, fecal samples were homogenized with 750 μL of chloroform: methanol 1:2 (v/v) and 10% deionized water, and incubated at 4 °C for 30 min. The samples were centrifuged after addition of 250 μL of chloroform and 350 μL of deionized water. Lipid in the lower organic phase after centrifugation was extracted twice. After that, the total extract was collected and dried in the SpeedVac under OH mode.

Agilent 1290 UPLC combined with a triple quadrupole/ion trap mass spectrometer (6500 Plus Qtrap; SCIEX) was used for FFAs analysis. Normal phase (NP)-HPLC with a Phenomenex Luna 3 μm-silica column (internal diameter 150×2.0 mm) was used for lipids separation. The conditions as follows; chloroform: methanol: ammonium hydroxide (89.5:10:0.5) was used as mobile phase A, and chloroform: methanol: ammonium hydroxide: water (55:39:0.5:5.5) was used as mobile phase B. D31-16:0 (Sigma-Aldrich) and d8-20:4 (Cayman Chemicals) was used as internal standards for FFAs quantitation.

## Statistical analysis

Statistical analysis was performed using SPSS (17.0, IBM, Armonk, NY, USA) and GraphPad Prism (San Diego, CA, USA). Data are presented as the mean ± SEM, and were compared using a repeated measure two-way analysis of variance (ANOVA), one-way ANOVA, or Student's *t-test*. Statistical significance was set at *p<0.05, **p<0.01, ***p<0.001.

## Acknowledgements

The author would like to appreciate Yanbian University for its support to Tumen River Scholars.

---

## Additional information

### Funding

| Funder | Grant reference number | Author |
|---|---|---|
| National Natural Science Foundation of China | 32260817 | Jin-Dan Kang |
| National Natural Science Foundation of China | 32260026 | Lin-Hu Quan |
| Changbai Mountain Talent Project of Jilin Province | 000007 | Lin-Hu Quan |
| Higher Education Discipline Innovation Project | D18012 | Lin-Hu Quan |
| Innovative and Entrepreneurial Talent in Jilin Province | 2023QN27 | Jin-Dan Kang |

The funders had no role in study design, data collection and interpretation, or the decision to submit the work for publication.

### Author contributions

Zhao-Bo Luo, Conceptualization, Data curation, Formal analysis, Investigation, Visualization, Writing - original draft, Writing - review and editing; Shengzhong Han, Data curation, Investigation, Visualization, Writing - original draft, Writing - review and editing; Xi-Jun Yin, Resources, Formal analysis, Methodology, Project administration, Writing - review and editing; Hongye Liu, Software, Investigation, Writing - original draft; Junxia Wang, Validation, Investigation, Visualization; Meifu Xuan, Formal analysis, Investigation, Visualization; Chunyun Hao, Danqi Wang, Yize Liu, Shuangyan Chang, Dongxu Li, Validation, Investigation; Kai Gao, Biaohu Quan, Investigation, Visualization; Huiling Li, Visualization, Writing - original draft; Lin-Hu Quan, Conceptualization, Data curation, Funding acquisition, Writing - original draft, Writing - review and editing; Jin-Dan Kang, Conceptualization, Formal analysis, Funding acquisition, Methodology, Writing - original draft, Writing - review and editing

### Author ORCIDs

Danqi Wang (iD) http://orcid.org/0009-0001-7656-9901
Lin-Hu Quan (iD) http://orcid.org/0000-0002-7195-8078

### Ethics

The animal study was approved by the Ethics Committee of Yanbian University (approval number SYXK2020-0009).

### Decision letter and Author response

Decision letter https://doi.org/10.7554/eLife.81858.sa1
Author response https://doi.org/10.7554/eLife.81858.sa2

---

## Additional files

### Supplementary files

- Supplementary file 1. Primers sequences used for real-time PCR.
- MDAR checklist

## Data availability

The raw reads of 16s rRNA gene sequences have been submitted to the NCBI BioSample database (Porcine data: PRJNA743164; Mice data: PRJNA743401). All sample metadata and intermediate analysis files are available at https://doi.org/10.57760/sciencedb.06767.

The following datasets were generated:

| Author(s) | Year | Dataset title | Dataset URL | Database and Identifier |
|---|---|---|---|---|
| Luo ZB | 2022 | Original data of Luo et al | https://doi.org/10.57760/sciencedb.06767 | Science Data Bank, 10.57760/sciencedb.06767 |
| Luo ZB, Han SZ, Yin XJ, Liu HY, Wang JX, Xuan MF, Hao CY, Wang DQ, Liu YZ, Chang SY, Gao K, Quan BH, Quan LH, Kang JD, Li DX, Li HI | 2021 | pig gut metagenome Raw sequence reads | https://www.ncbi.nlm.nih.gov/bioproject/?term=PRJNA743164 | NCBI BioProject, PRJNA743164 |
| Luo ZB, Han SZ, Yin XJ, Liu HY, Wang JX, Xuan MF, Hao CY, Wang DQ, Liu YZ, Chang SY, Gao K, Quan BH, Quan LH, Kang JD, Li DX, Li HI | 2021 | mice gut metagenome Raw sequence reads | https://www.ncbi.nlm.nih.gov/bioproject/?term=PRJNA743401 | NCBI BioProject, PRJNA743401 |

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
