## [Editor Report]

This study highlights how the deletion of the MSTN gene in pigs affects the gut microbiota and leads to changes in skeletal muscle growth and function. By transplanting the remodeled gut microbiota from MSTN-deleted pigs to mice, the authors demonstrate the selective hypertrophy of fast-twitch glycolytic muscles. Additionally, valeric acid, a microbial metabolite produced in the gut, promotes skeletal muscle growth by activating the Akt/mTOR pathway via the SCFA receptor GPR43 and has potential therapeutic implications for muscle diseases such as muscular dystrophy and sarcopenia.

---

## [Decision Letter]

**Decision letter after peer review:**

Thank you for submitting your article "Myostatin gene deletion alters gut microbiota stimulating fast-twitch glycolytic muscle growth" for consideration by *eLife*. Your article has been reviewed by 3 peer reviewers, one of whom is a member of our Board of Reviewing Editors, and the evaluation has been overseen by a Reviewing Editor and Wendy Garrett as the Senior Editor. The reviewers have opted to remain anonymous.

Essential revisions:

Though there was unanimous interest in this study, reviewers cite concerns with the level of novelty and mechanistic understanding of the phenotypes reported. Additional experiments are requested. Suggestions to improve the manuscript include demonstrating a causal mechanism between MSTN KO and alterations to the microbiome, or mechanistically determining why the effect of valerate is distinct from acetate, propionate, and butyrate. Additional recommendations for revisions are noted below. Importantly, the manuscript should be thoroughly reviewed for English grammar and syntax.

*Reviewer #1 (Recommendations for the authors):*

There are several issues with the study:

Is the concentration of valerate in the daily oral gavage consistent with the concentration of valerate naturally in the FMT mice?

Is there evidence of increased valeric acid in the MSTN KO pigs?

The significance of the functional metagenomic analysis in the mice (Fig3E) is unclear. First, please display values across all individual mice; as-is, it appears that the values are somehow averaged across individuals. Second, it is strange that the enrichment values appear binary (i.e. either -0.6 or +0.6, but no values in-between). Third, the plot only displays K0 numbers, which are not human-readable. If including this plot in the main text, please include the description of each K0 number and some kind of hierarchical groupings of K0 numbers so that the trend is more apparent, if there is a trend.

*Reviewer #2 (Recommendations for the authors):*

A better title is needed. Maybe, "Fecal transplant from myostatin KO animals positively impacts the gut-muscle axis"?

Extensive editing for English syntax and grammar is needed.

Body weight (Supp Figure 2a) was higher in KO-M starting at week 5 but isn't mentioned in the manuscript. Are soleus, gastroc, and EDL masses different when compared between groups when each muscle is normalized to body weight? All muscle measures should be divided by body weight, and then compared between groups.

Where is the data for "food intake, physical activity, energy intake, or absorbed energy"? It's not in Figure 2 or Supplemental Figure 2a-e, as cited in the manuscript.

PCA analysis for pig and mice data should be performed together to see if microbiota from colonized mice clusters with their respective pig donors-if so, that would add strength to the idea that the microbiome was successfully transplanted, and may play a role on muscle.

Line 45 should read, "Myostatin (MSTN) deletion, a key factor *that regulates* muscle growth,…"

Line 46: *pig* should come before MSTN.

Line 47: showed *an* increase.

Line 67: should read, "increases *morbidity* and mortality *risk*".

Line 69: are the references for this sentence in the previous sentence?

Line 77: should read, *and participates*.

Line 85: delete *the* before GLUT1.

Line 89: delete *in*.

Line 95: should read, *including* Alistipes and Veillonella,". Delete "respectively".

Line 96: Add a comma after "2022".

Line 100: *improve*, not "improving"; *control*, not "controlling", *regulate*, not "regulating".

Lines 103 – 104: "SCFAs play a vital role in skeletal 104 muscle mass maintenance (Lv et al., 2021; Chen et al., 2022)". These are association-based studies. A causative role for SCFAs on muscle mass and strength is provided in Lahiri et al., which is cited later in the test. Lahiri should be cited here.

Line 107 – 118: This paragraph seems out of place here; parts of it can be deleted and/or moved to the Discussion.

Line 133: "We found that *MSTN−/− pigs*, delete *those*.

Line 150: *diversity*, not "abundance".

Figure 2B: EDL is more glycolytic in terms of Tyle II fibers than Gastrocnemius-what is the rationale for why EDL muscle mass in Figure 2B wasn't increased when compared with WT?

Figure 2C: These images aren't great, barely redder in KO-M.

Line 192: Romboutsia is the genus-its corresponding family, Peptostreptococcaceae was increased, and should be mentioned in the text. In other words, Romboutsia is not found at the genus, family, and order taxonomic levels.

Line 225: What were the body weight, food intake, and physical activity in VA-treated mice when compared with controls? Muscle mass should be normalized to body weight and compared between groups.

Line 244: "but had no effect on food intake, physical activity, energy intake, or absorbed energy in mice (Figure 6—figure supplement 4A-E)" That data is not in Figure 6 or Supplementary 4A-E.

Line 267: *microbial*, not "microbioal".

*Reviewer #3 (Recommendations for the authors):*

1. In lines 146-149, where the authors describe α-diversity values, it is unclear what ACE, Chao 1, Shannon, and Simpson indices refer to. Therefore, a clarification is required. Also, the authors should explain why this result is meaningful.

2. There are some grammatical errors (for example, line 161 should be 'transplanted' not 'translated'). Proofreading of the manuscript is essential.

3. Figure 3E depicts deferentially enriched pathways related to metabolite synthesis between mice transplanted with WT or MSTN KO microbiome. The authors need to clarify what each number stands for, and the overall interpretation of this analysis. The author points out K05349 and K01952 without explaining their implications, and it is unclear why these metabolites are not further pursued.

4. The heatmap in Figure 4D is quite variable. The author should comment on this instead of concluding that the 'heatmap confirmed the differences in SCFAs…'.

5. The link between valeric acid and GPR43 is not well established. Can the authors perform knockdown experiments for GPR43 to confirm that valeric acid is acting through it? Just observing changes in expression is not sufficient to describe the mechanism.

6. Can the authors comment on the implications of the increased length of the small intestine in Figure 6I? At this point, such observations are merely descriptive.

[Editors’ note: further revisions were suggested prior to acceptance, as described below.]

Thank you for resubmitting your work entitled "Fecal transplant from myostatin deletion pigs positively impacts the gut-muscle axis" for further consideration by *eLife*. Your revised article has been evaluated by Wendy Garrett (Senior Editor) and a Reviewing Editor.

The manuscript has been improved but there are some remaining issues that need to be addressed, as outlined by Reviewer #3 below.

*Reviewer #3 (Recommendations for the authors):*

The authors have performed additional experiments and made clarifications to support their studies. However, the link of MSTN to intestinal barrier function is still weak. It is recommended that the authors perform rescue experiments by putting back MSTN specifically in the intestinal smooth muscle to demonstrate that not only are the intestinal barrier defects rescued but so is the skeletal muscle phenotype.

Alternatively, conditional MSTN KO pigs can be used to delete MSTN specifically in the intestine and show the effects that altered gut microbiota has on other tissues such as the skeletal muscle.

Currently, the resulting muscle phenotype is likely due to cell-autonomous changes in the MSTN KO muscle cells, thus it may not be related to the intestinal smooth muscle phenotype. This is the major caveat of this paper.

---

## [Author Response]

Essential revisions:Though there was unanimous interest in this study, reviewers cite concerns with the level of novelty and mechanistic understanding of the phenotypes reported. Additional experiments are requested. Suggestions to improve the manuscript include demonstrating a causal mechanism between MSTN KO and alterations to the microbiome, or mechanistically determining why the effect of valerate is distinct from acetate, propionate, and butyrate. Additional recommendations for revisions are noted below. Importantly, the manuscript should be thoroughly reviewed for English grammar and syntax.

Thanks to the reviewers for this exciting suggestion. The causal mechanism between MSTN KO and alterations to the microbiota is indeed a valuable study. For this point, we conducted additional experiments in the revised manuscript. Considering that intestinal structure and barrier function affect intestinal microbiota (Gallo et al., 2012; Seganfredo et al., 2017), and MLCK/MLC is an important pathway regulating intestinal barrier function, therefore, we detected the expression of MLCK and the phosphorylation of MLC in the intestine of MSTNKO pigs by western blot, and observed low levels of MLCK and phosphorylation of MLC. Furthermore, inhibition of MSTN in IPEC-J2 cells increased the expression of intestinal tight junction factors ZO-1 and occludin. With the changes in the intestinal muscularis and plicae, and the improvement of intestinal tight junction by MLCK/MLC in MSTN KO pigs, these findings provide strong evidence that the structure and barrier function of intestine is implicated in the alterations of the microbiota by MSTN deletion. These results showed in Figure 2.

In addition, we thank the reviewer for drawing attention to the difference between valeric acid and other SCFAs. To address this comment, we added new data that both of valeric acid and butyric acid can activate Akt/mTOR pathway through GPR43, but not acetic acid and propionic. Importantly, our data indicated that only valeric acid can promote the expression of myogenic differentiation factors, MyoD and MyoG. We further demonstrated the mechanism by which valeric acid promotes skeletal muscle mass through GPR43 in in vivo and in vitro. The inhibition of GPR43 in mice and the knock down of GPR43 in myoblasts were found to prevent valeric acid-induced Akt/mTOR pathway activation. These results showed in Figure 8.

And, we corrected the grammatical errors throughout the manuscript.

Reference

Gallo RL, Hooper LV. Epithelial antimicrobial defence of the skin and intestine. 2012. *Nat Rev Immunol* 12, 503-16. DOI: https://doi:10.1038/nri3228, PMID: 22728527

Seganfredo FB, Blume CA, Moehlecke M, Giongo A, Casagrande DS, Spolidoro JVN, Padoin AV, Schaan BD, Mottin CC. 2017. Weight-loss interventions and gut microbiota changes in overweight and obese patients: a systematic review. *Obes Rev* 18, 832-851. DOI: https://doi:10.1111/obr.12541, PMID: 28524627

Reviewer #1 (Recommendations for the authors):There are several issues with the study:Is the concentration of valerate in the daily oral gavage consistent with the concentration of valerate naturally in the FMT mice?

The valeric acid detected in the intestinal contents was 1μmol/g and the concentration of oral valeric acid in this study was based on the concentration of SCFAs used in other studies (Lin et al., 2012; Li et al., 2018; Lanza et al., 2021).

Is there evidence of increased valeric acid in the MSTN KO pigs?

Thanks to the reviewer for raising this question, because it's really an overlooked point. We analyzed the content of valeric acid in feces of MSTN KO pigs and WT pigs, and found that valeric acid was significantly higher in MSTN KO pigs than WT pigs.

The significance of the functional metagenomic analysis in the mice (Fig3E) is unclear. First, please display values across all individual mice; as-is, it appears that the values are somehow averaged across individuals. Second, it is strange that the enrichment values appear binary (i.e. either -0.6 or +0.6, but no values in-between). Third, the plot only displays K0 numbers, which are not human-readable. If including this plot in the main text, please include the description of each K0 number and some kind of hierarchical groupings of K0 numbers so that the trend is more apparent, if there is a trend.

We agree with the reviewer's suggestion and have made a new heatmap, which shows the values for all individual mice and depicts each K0 number. And we made changes in results part. It was shown in Figure 4E.Reference

Lin HV, Frassetto A, Kowalik EJ Jr, Nawrocki AR, Lu MM, Kosinski JR, Hubert JA, Szeto D, Yao X, Forrest G, Marsh DJ. 2012. Butyrate and propionate protect against diet-induced obesity and regulate gut hormones via free fatty acid receptor 3-independent mechanisms. *PLoS One* 7, e35240. DOI: https://doi:10.1371/journal.pone.0035240, PMID: 22506074

Li Z, Yi CX, Katiraei S, Kooijman S, Zhou E, Chung CK, Gao Y, van den Heuvel JK, Meijer OC, Berbée JFP, Heijink M, Giera M, Willems van Dijk K, Groen AK, Rensen PCN, Wang Y. 2018. Butyrate reduces appetite and activates brown adipose tissue via the gut-brain neural circuit. *Gut* 67, 1269-1279. DOI: https://doi:10.1136/gutjnl-2017-314050, PMID: 29101261

Lanza M, Filippone A, Ardizzone A, Casili G, Paterniti I, Esposito E, Campolo M. 2021. SCFA Treatment Alleviates Pathological Signs of Migraine and Related Intestinal Alterations in a Mouse Model of NTG-Induced Migraine. *Cells* 14, 2756. DOI: https://doi:10.3390/cells10102756, PMID: 34685736

Reviewer #2 (Recommendations for the authors):A better title is needed. Maybe, "Fecal transplant from myostatin KO animals positively impacts the gut-muscle axis"?

We thank the reviewer for their thorough analysis of our experimental design and we have changed the title.

Extensive editing for English syntax and grammar is needed.

Yes, we have corrected the grammatical errors.

Body weight (Supp Figure 2a) was higher in KO-M starting at week 5 but isn't mentioned in the manuscript. Are soleus, gastroc, and EDL masses different when compared between groups when each muscle is normalized to body weight? All muscle measures should be divided by body weight, and then compared between groups.

We reuploaded the original data corresponding to the results. All muscles tended to rise compared to the control group. Body weight normalization analysis of total skeletal muscle mass showed that the KO-M group was significantly higher than the WT-M group (see Author response table 1), so we believed that skeletal muscle was enlarged overall.

**Author response table 1. sa2table1:** 

	Body Weight(g)	GA(mg)	SOL(mg)	EDL(mg)	TA(mg)	GA/BW(%)	SOL/BW(%)	EDL/BW(%)	TA/BW(%)	Muscle/BW(%)
WT-M	25.65	287.73	19.16	20.93	94.72	11.22	0.75	0.82	3.69	16.47
KO-M	26.7	305.77	20.22	22.05	105.42	11.45	0.76	0.83	3.94	16.98
TTEST	0.02	0.02	0.22	0.28	0.05	0.37	0.70	0.79	0.11	0.04

Where is the data for "food intake, physical activity, energy intake, or absorbed energy"? It's not in Figure 2 or Supplemental Figure 2a-e, as cited in the manuscript.

I think this is a mistake. The supplementary materials cannot be displayed normally in the system, and I have made a new upload.

PCA analysis for pig and mice data should be performed together to see if microbiota from colonized mice clusters with their respective pig donors-if so, that would add strength to the idea that the microbiome was successfully transplanted, and may play a role on muscle.

PCA analysis of pigs and mice showed that the distribution of microbiota of recipient mice was closer to that of donor pigs, and the distribution of KO-M microbiota was more similar to that of MSTN KO pigs. It is shown in Author response image 1.

**Author response image 1. sa2fig1:** 

Line 45 should read, "Myostatin (MSTN) deletion, a key factor that regulates muscle growth,…"

Yes, I have changed it in the revised manuscript.

Line 46: pig should come before MSTN.

Yes, I did it.

Line 47: showed an increase.

Yes, I did it.

Line 67: should read, "increases morbidity and mortality risk".

Yes, I did it.

Line 69: are the references for this sentence in the previous sentence?

Yes, I have added the reference here.

Line 77: should read, and participates.

Yes, I did it.

Line 85: delete the before GLUT1.

Yes, I did it.

Line 89: delete in.

Yes, I did it.

Line 95: should read, including Alistipes and Veillonella,". Delete "respectively".

Yes, I did it.

Line 96: Add a comma after "2022".

Yes, I did it.

Line 100: improve, not "improving"; control, not "controlling", regulate, not "regulating".

Yes, I did it.

Lines 103 – 104: "SCFAs play a vital role in skeletal 104 muscle mass maintenance (Lv et al., 2021; Chen et al., 2022)". These are association-based studies. A causative role for SCFAs on muscle mass and strength is provided in Lahiri et al., which is cited later in the test. Lahiri should be cited here.

Yes, I did it.

Line 107 – 118: This paragraph seems out of place here; parts of it can be deleted and/or moved to the Discussion.

Yes, I did it.

Line 133: "We found that MSTN−/− pigs, delete those.

I have deleted it.

Line 150: diversity, not "abundance".

Yes, I changed it.

Figure 2B: EDL is more glycolytic in terms of Tyle II fibers than Gastrocnemius-what is the rationale for why EDL muscle mass in Figure 2B wasn't increased when compared with WT?

Thank you for your professional comments and we agree with you. Gastrocnemius is a mixed muscle fiber, and EDL is a glycolytic muscle fiber. Actually, weight of EDL showed a tendency to increase, but there was no significant difference in our data. At this point, we think it is due to that the mass of EDL is smaller when compared with gastrocnemius. However, we analyzed the glycolysis ability of EDL and found that the KO-M group was significantly improved (Author response image 2).

Figure 2C: These images aren't great, barely redder in KO-M.

Yes, we changed them. Figure 3C.

Line 192: Romboutsia is the genus-its corresponding family, Peptostreptococcaceae was increased, and should be mentioned in the text. In other words, Romboutsia is not found at the genus, family, and order taxonomic levels.

We thank the reviewer for pointing this and we have changed it.

Line 225: What were the body weight, food intake, and physical activity in VA-treated mice when compared with controls? Muscle mass should be normalized to body weight and compared between groups.

Yes, the results of food intake, and physical activity in Val-treated mice were in supplementary Figure 4.

Body weight normalization analysis of total skeletal muscle mass showed that the Val-treated group was significantly higher than the controls (see Author response table 2). All skeletal muscles of the Val-treated mice showed a tendency to increase compared to the control group.

**Author response table 2. sa2table2:** 

	Body Weight(g)	GA(mg)	SOL(mg)	EDL(mg)	TA(mg)	GA/BW(%)	SOL/BW(%)	EDL/BW(%)	TA/BW(%)	Muscle/BW(%)
WT-M	23.23	264.24	13.31	17.32	77.09	11.38	0.57	0.74	3.31	16.0
KO-M	25.12	288.59	14.48	19.34	86.19	11.49	0.57	0.77	3.43	16.26
TTEST	0.01	0.02	0.12	0.05	0.05	0.64	0.75	0.41	0.38	0.04

Line 244: "but had no effect on food intake, physical activity, energy intake, or absorbed energy in mice (Figure 6—figure supplement 4A-E)" That data is not in Figure 6 or Supplementary 4A-E.

I think this is a mistake. The supplementary materials cannot be displayed normally in the system, and I have made a new upload.

Line 267: microbial, not "microbioal".

Yes, I have changed it.

Reviewer #3 (Recommendations for the authors):1. In lines 146-149, where the authors describe α-diversity values, it is unclear what ACE, Chao 1, Shannon, and Simpson indices refer to. Therefore, a clarification is required. Also, the authors should explain why this result is meaningful.

Yes, we made some changes in Results sections. We thank the reviewer for pointing this out. Α diversity reflects species richness, evenness and sequencing depth. ACE and Chao1 mainly estimate the number of species, known as richness. Shannon index reflects diversity. Simpson reflects richness and evenness. ACE decreased significantly after MSTN KO, indicating that the species of microbiota decreased in MSTN KO pigs. These revisions were added in the Results section.

2. There are some grammatical errors (for example, line 161 should be 'transplanted' not 'translated'). Proofreading of the manuscript is essential.

Yes, I have changed it and corrected the grammatical errors.

3. Figure 3E depicts deferentially enriched pathways related to metabolite synthesis between mice transplanted with WT or MSTN KO microbiome. The authors need to clarify what each number stands for, and the overall interpretation of this analysis. The author points out K05349 and K01952 without explaining their implications, and it is unclear why these metabolites are not further pursued.

We agree with the reviewer's suggestion and have made a new heatmap (see Figure 4E), which shows the values for all individual mice and depicts each K0 number. The two up-regulated pathways we focused on were related to the biosynthesis of secondary metabolites, which suggested that changes in microbiota structure promoted the synthesis of secondary metabolites.

4. The heatmap in Figure 4D is quite variable. The author should comment on this instead of concluding that the 'heatmap confirmed the differences in SCFAs…'.

I made some changes in Results section. Heat map analysis showed that SCFAs content in KO-M group and WT-M group was significantly different on the whole level. SCFAs in KO-M were significantly upregulated in four out of seven mice, and the variables of heat map may be caused by individual differences of mice.

5. The link between valeric acid and GPR43 is not well established. Can the authors perform knockdown experiments for GPR43 to confirm that valeric acid is acting through it? Just observing changes in expression is not sufficient to describe the mechanism.

Thanks for the reviewer's constructive suggestion. We further verified the mechanism by which valeric acid promotes skeletal muscle mass through in vitro and in vivo experiments. Mice were oral treated with valeric acid and GPR43-specific inhibitor GLPG0974, and it was found that valeric acid did not promote skeletal muscle mass in the presence of GPR43 inhibitor. In addition, GPR43 was knocked down in C2C12 and it was found that valeric acid did not activate Akt and mTOR. These results were shown in Figure 8.

6. Can the authors comment on the implications of the increased length of the small intestine in Figure 6I? At this point, such observations are merely descriptive.

Thank you for such an important comment. We have added more descriptions of this.

[Editors’ note: further revisions were suggested prior to acceptance, as described below.]

Reviewer #3 (Recommendations for the authors):The authors have performed additional experiments and made clarifications to support their studies. However, the link of MSTN to intestinal barrier function is still weak. It is recommended that the authors perform rescue experiments by putting back MSTN specifically in the intestinal smooth muscle to demonstrate that not only are the intestinal barrier defects rescued but so is the skeletal muscle phenotype.Alternatively, conditional MSTN KO pigs can be used to delete MSTN specifically in the intestine and show the effects that altered gut microbiota has on other tissues such as the skeletal muscle.

Thank you very much for your detailed evaluation of our research. We appreciate the time and effort you put into reviewing our work.

Based on your suggestions, we have conducted new experiments. To clarify the association between MSTN and intestinal barrier function, we conducted a rescue experiment by adding recombinant MSTN (Rc-MSTN) protein to IPEC-J2 cells after MSTN knockout (MSTN-KD). The Western blot analysis revealed that the expression of MLCK and p-MLC, which were decreased by MSTN-KD, were restored upon the addition of Rc-MSTN. Furthermore, the mRNA levels of *TJP1* and *OCLN* were up-regulated in MSTN-KD, whereas their expression decreased following the addition of Rc-MSTN. We found that the MSTN KO in pig altered intestinal barrier function. Therefore, we utilized the commonly used IPEC-J2 cell line, which is associated with intestinal (smooth muscle) barrier function and tight junction, to conduct our experiments (PMID: 22074860; PMID: 36366564; PMID: 35384371).

The TALEN plasmid we used was consistent with these used for producing the MSTN^-/-^ pigs. Left TALEN site: TTCAAATCCTCAGTAAACTT, Right TALEN site: CTCCTAACATTAGCAAAGA. Rc-MSTN from RD system and used for 500 ng/ml, 24 h.

**Author response image 3. sa2fig3:** 

Currently, the resulting muscle phenotype is likely due to cell-autonomous changes in the MSTN KO muscle cells, thus it may not be related to the intestinal smooth muscle phenotype. This is the major caveat of this paper.

We highly appreciate the opinions and suggestions provided by the reviewer, which are very valuable for improving our research. Actually, MSTN-KO pigs exhibit stronger muscle hypertrophy and fiber enlargement, displaying a clear "double muscle" phenotype involving a series of muscles such as the longissimus dorsi, biceps femoris, and semitendinosus (the average myofiber sizes of longissimus dorsi were 2177.7 μm^2^, significantly larger than WT in 1564.3 μm^2^, 39.2% increased). However, the degree of increased muscle mass and fiber enlargement observed in mice after fecal microbiota transplantation was limited, only gastrocnemius muscle mass was significantly increased (the average myofiber sizes of GA were 1634.5 μm^2^, significantly larger than WT-M in 1453.7 μm^2^, 12.4% increased), and the phenotype of increased skeletal muscle mass in mice was not as significant as that observed in MSTN-KO pigs. In our study, although the mice did not fully exhibit the same degree of skeletal muscle hypertrophy as the MSTN-KO pigs, the microbiota transplantation did increase skeletal muscle mass and promoted muscle fiber type IIb consistent with the MSTN-KO pigs. This suggests that the gut microbiota does play a role in this process. We have made some changes in the Discussion section. Line 278 to 285.

We concur with the reviewer's perspective that MSTN KO leads to autonomous alterations in muscle cells. Several studies have shown that MSTN inhibition directly stimulates the proliferation and differentiation of myoblast (PMID: 32156541; PMID: 28955860). As such, we posit that the cell-autonomous changes induced by the MSTN gene in muscle cells may be its primary pathway. Furthermore, the effect of gene-mediated changes in gut microbiota on skeletal muscle is also a novel pathway worth considering. Indeed, our study results undeniably indicate that the altered gut microbiota due to MSTN gene deficiency also partially contributes to the impact on skeletal muscle.

Although there have been relevant reports proving that gut specific gene knockout affects intestinal microbiota composition, there is no study on global MSTN gene knockout changes intestinal microbiota through the gut. Therefore, this study determined the pathway by which the MSTN gene affects skeletal muscle development through changing intestinal microflora under the condition that adult MSTN KO pigs have a "double muscle" phenotype. The experimental plan proposed by the reviewer for gut-specific MSTN KO in pigs is valuable. In future studies, we will consider and refer to this method to investigate the effects of gut-specific MSTN on microbiota and skeletal muscle.

In conclusion, we believe that the intestinal microflora partially participates in the skeletal muscle increase caused by MSTN deficiency, and the results of the mouse fecal microbiota transplantation confirm that gut microbiota is one of the links between the MSTN gene and skeletal muscle.

We hope that these revisions have addressed your concerns and improved the quality of our manuscript.

Brosnahan AJ, Brown DR. 2012. Porcine IPEC-J2 intestinal epithelial cells in microbiological investigations. *Vet Microbiol* 156, 229-237. DOI: https://doi:10.1016/j.vetmic.2011.10.017, PMID: 22074860

Cornelius V, Droessler L, Boehm E, Amasheh S. 2022. Concerted action of berberine in the porcine intestinal epithelial model IPEC-J2: Effects on tight junctions and apoptosis. *Physiol Rep* 10, e15237. DOI: https://doi: 10.14814/phy2.15237. PMID: 35384371

Ge L, Dong X, Gong X, Kang J, Zhang Y, Quan F. 2020. Mutation in myostatin 3'UTR promotes C2C12 myoblast proliferation and differentiation by blocking the translation of MSTN. *Int J Biol Macromol* 54, 634-643. DOI: https://doi: 10.1016/j.ijbiomac.2020.03.043. PMID: 32156541

Liu X, Wang Y, Han C, Li Q, Hou X, Song Q, Zhou S, Li H. 2022. TGF-β from the Porcine Intestinal Cell Line IPEC-J2 Induced by Porcine Circovirus 2 Increases the Frequency of Treg Cells via the Activation of ERK (in CD4^+^ T Cells) and NF-κB (in IPEC-J2). *Viruses* 14, 2466. DOI: https://doi: 10.3390/v14112466. PMID: 36366564

Pèrié L, Parenté A, Brun C, Magnol L, Pélissier P, Blanquet V. 2016. Enhancement of C2C12 myoblast proliferation and differentiation by GASP-2, a myostatin inhibitor. *Biochem Biophys Rep* 6, 39-46. DOI: https://doi:10.1016/j.bbrep.2016.03.001. PMID: 28955860